# Exploring the Potential Biocontrol Isolates of *Trichoderma asperellum* for Management of Collar Rot Disease in Tomato

C. Shanmugaraj [1], Deeba Kamil [1,*], Aditi Kundu [2], Praveen Kumar Singh [3], Amrita Das [1], Zakir Hussain [4], Robin Gogoi [1], P. R. Shashank [5], R. Gangaraj [1] and M. Chaithra [1]

1    Division of Plant Pathology, ICAR-Indian Agricultural Research Institute, New Delhi 110012, India; spcshanmugaraj@gmail.com (C.S.); amrita.das@icar.org.in (A.D.); rgogoi@iari.res.in (R.G.); gangaeverest.123@gmail.com (R.G.); chaithram06@gmail.com (M.C.)
2    Division of Agricultural Chemicals, ICAR-Indian Agricultural Research Institute, New Delhi 110012, India; aditi@iari.res.in
3    Division of Centre for Protected Cultivation Technology (CPCT), ICAR-Indian Agricultural Research Institute, New Delhi 110012, India; pksingh128@iari.res.in
4    Division of Vegetable Science, ICAR-Indian Agricultural Research Institute, New Delhi 110012, India; zakir.hussain1@icar.gov.in
5    Division of Entomology, ICAR-Indian Agricultural Research Institute, New Delhi 110012, India; shashank@iari.res.in
*    Correspondence: deeba.kamil@icar.gov.in

**Abstract:** Bio-control agents are the best alternative to chemicals for the successful management of plant diseases. Among them, *Trichoderma* is commonly used as a biological control agent in plant disease management due to its ability to suppress soil-borne plant pathogens. In the present study, 20 *Trichoderma asperellum* isolates were collected from different geographical locations and confirmed using morphological characteristics and molecular phylogenetic inferences based on combined ITS and β-tubulin sequences. All twenty isolates were screened for their antagonism against the collar rot pathogen under in vitro and in planta conditions. The isolates were evaluated through dual culture and volatile methods in an in vitro study. Isolate A10 inhibited the test pathogen *Agroathelia rolfsii* at 94.66% in a dual culture assay and 70.95% in a volatile assay, followed by the isolates A11 and A17, which recorded 82.64% and 81.19% in dual culture assay and 63.75% and 68.27% in the volatile assay respectively. An in planta study was conducted under greenhouse conditions in tomato var. pusa ruby by pre- and post-inoculation of *T. asperellum* isolates in the *A. rolfsii* infected soil to evaluate their antagonistic potential against the disease. The A10 isolate was found effective under both pre- and post-inoculation conditions, with a disease inhibition percent of 86.17 and 80.60, respectively, followed by the isolates A11 and A17, which exhibited inhibition of 77.80% and 75.00% in pre-inoculation and 72.22% and 69.44% in post-inoculation, respectively. Further, biochemical analysis was conducted to determine the specific activity of hydrolytic enzymes produced by *T. asperellum* during interaction with *A. rolfsii*. We found that isolate A10 produces more hydrolytic enzymes with the specific activity of 174.68 IU/mg of β-1,3 glucanase, 183.48 IU/mg of β-1, 4 glucanase, 106.06 IU/mg of protease, followed by isolate A17, A11 respectively. In GC-MS analysis, we observed maximum anti-microbial volatile organic compounds from the isolate A10, including 2H-Pyran-2-one (17.39%), which was found to be most abundant, followed by dienolactone (8.43%), α-pyrone (2.19%), and harziandione (0.24%) respective retention time of 33.48, 33.85, 33.39, and 64.23 min, respectively, compared to other isolates. In the TLC assay, we observed that a greater number of bands were produced by the A10 and A17 isolates in the Hexane: Ethyl Acetate (1:1) solvent system than in the 9:1 solvent system, which represents the presence of major metabolites in the ethyl acetate extract.

**Keywords:** bio-control; bio-efficacy; GC-MS; hydrolytic enzymes; *Agroathelia rolfsii*; secondary metabolites; *Trichoderma asperellum*

## 1. Introduction

*Agroathelia rolfsii* (Sacc.) Redhead and Mullineux (Agaricomycetes: Amylocorticiales) is a destructive, necrotrophic, soil-borne plant pathogen that causes collar rot or southern blight disease [1]. It was first reported in Florida in 1892 by Rolfs on tomatoes. They have an extremely wide host range that includes more than 600 plant species across 100 plant families, such as groundnut, green bean, lima bean, onion, garden bean, pepper, potato, sweet potato, tomato, and watermelon [2–4], causing huge losses worldwide. It causes significant losses in various crops in India, including peanuts, soybeans, tomatoes, peppers, and many others. In peanut and tomato crops, the losses due to *A. rolfsii* in India have been estimated to be around 30–40% of the total crop yield [5–7]. Management is not successful in the field because of the soil-borne nature of the pathogen.

Moreover, it produces prolific growth and can produce persistent sclerotia, which plays a key role in the disease cycle. Like many other soil-borne fungal diseases, *A. rolfsii* continues to be a difficult pathogen to control due to its wide host range, abundant persistent sclerotia, and genetic variability among populations [8]. Biological control is an environmentally sound and effective means of managing plant diseases. Recently, the use of potential biocontrol agents has gained importance for the management of pests and diseases because of their cost-effectiveness, sustainability, and eco-friendly nature.

The genus *Trichoderma* is a group of fungi that is well-known for their biocontrol potential against a range of plant pathogens, including *A. rolfsii* [9–12]. *Trichoderma* spp. are natural antagonists that can compete with plant pathogens via various mechanisms, *viz.*, space and nutrients, production of enzymes, and secondary metabolites that can inhibit or kill the pathogens [13,14] and confrontation through mycoparasitism or either by inducing resistance and plant defense reactions [15–17].

However, *Trichoderma asperellum* (Sordariomycetes; Hypocreaceae) possesses a dual activity of both biocontrol and plant growth-promoting properties [18–20]. The colonization of the *Arabidopsis* root by *T. asperellum* SKT-1 and its culture filtrate elicit an ISR (Induced Systemic Resistance) against *Pseudomonas syringae* pv. *tomato* DC3000 through the increased expression of JA (Jasmonic Acid)/ET (Ethylene) and SA (Salicylic Acid) inducing genes [21]. Similar to Plant Growth Promoting Rhizobacteria (PGPR), *T. asperellum*- ISR is activated by the JA/ET signaling pathway [22]. For instance, *T. asperellum* was used by a number of nations to stop crown rot, root rot, and damping off [23]. Swollenin production is induced in plants as a result of *T. asperellum* colonization of the roots to provide a local defense [24]. The ethylene and $H_2O_2$-mediated plant defense responses against *Rhizoctonia solani* attacks were induced by the synthesis of *T. asperellum* xylanases in plants [25].

There were many secondary metabolites like 6-pentyl-2H-pyran-2-one (6-PP), 2,4-ditert-butyl phenol, propenyl phenyl methyl ester, heptanes, viridin, and harzianolides were reported from *Trichoderma* spp. [26]. More importantly, metabolites, like 6-pentyl-2H-pyran-2-one, have antifungal activity against various plant pathogens [27]. The isolates of *T. asperellum* were known to produce several hydrolytic enzymes like cellulase, β-1,3-glucanases, β-1,4-glucanases, chitinase, and protease with antifungal ability by degrading the cell wall components of the various plant pathogens [28–31].

Therefore, the present study was carried out to evaluate *T. asperellum* isolates for their antagonistic activity against the collar rot pathogen under in vitro and in planta conditions and to characterize the bioactive compounds produced by this bio-control agent. This information will facilitate extensive applications of formulation in the field of bio-control in the future for the successful management of collar rot disease.

## 2. Materials and Methods

### 2.1. Collection and Isolation of the Fungus

In 2022, from March to October, 20 soil and rootsamples were collected from the nine states of India (New Delhi, Andhra Pradesh, Rajasthan, West Bengal, Gujarat, Karnataka, Maharashtra, Tamil Nadu, and Uttar Pradesh). Isolation of the potential fungal biocontrol isolates from soil samples was performed on potato dextrose agar (PDA) according to

the procedure described by [32]. Plates were incubated at 28 ± 2 °C for 5–7 days. The *Trichoderma* growth was obtained after 5–7 days and further purified by sub-culturing. The purified fungal isolates were maintained on PDA slants at 4°C. All the twenty obtained isolates were deposited in the Indian Type Culture Collection (ITCC) at the Indian Agricultural Research Institute, New Delhi. A highly virulent isolate *A. rolfsii* (Sr38), obtained during our previous work (ITCC Code: 8665, Accession number: OR192927), was used to evaluate the biocontrol potential of the obtained twenty *T. asperellum* isolates.

## 2.2. Morphological Observations

Morphological identification of the twenty potential biocontrol *T. asperellum* isolates was performed under the compound microscope along with camera attachment (ProgRes capture pro2.7- JENOPTIK). Isolates were identified based on microscopic features, including conidiophores branching, the shape and size of the phialides, and the conidia.

## 2.3. Molecular Identification and Phylogenetic Analysis

DNA from all the twenty isolates was extracted using a CTAB (Cetyl Trimethyl Ammonium Bromide), procedure given by Cullings [33]. The amplification of the ITS region was carried out using universal primers ITS1 5′-TCCGTAGGTGAACCTGCGG-3′ and ITS4 5′-TCCTCCGCTTATTGATATGC-3′ [34], B-tubf1-F 5′-CAGCTCGAGCGTATGAACGTCT-3′and B-tubr1-R 5′-TGTACCAATGCAAGAAAGCCTT-3′ primers were used for the amplification of β-tubulin region [35]. A total volume of 25 μL PCR mixture consisted of 12.5 μL DreamTaq Green PCR master mix (Thermo Scientific, Pune, India) (including 0.25 mM each of dNTP, 2mM MgCl$_2$, and Taq DNA polymerase), 9.5 μL nuclease-free water, 1 μL (10 Pmol/ μL of each forward and reverse primer), and 1 μL (100 ng/ μL) of fungal DNA. PCR was performed with initial denaturation for 3 min at 94 °C, 30 cycles of denaturation at 94 °C for 30 s, annealing temperature at 57 °C for 1 min, primer extension at 72 °C for 2 min, and final primer extension at 72 °C for 5 min for the ITS region. The PCR for β-tubulin was performed with initial denaturation for 3 min at 94 °C, 30 cycles of denaturation at 94 °C for 30 s, annealing temperature at 61 °C for 40 s, primer extension at 72 °C for 30 s, and final primer extension at 72 °C for 7 min. The electrophoresis was performed using 1.2% agarose gel with 0.5 mg/μL ethidium bromide in 1xTAE buffer (100 V, 400 mA for 30 min) to visualize the PCR products. The 1 kb DNA marker (Thermo Fisher Scientific, Waltham, MA, USA) was used to estimate the size of the PCR products, and further, the samples were sequenced through outsourcing. The NCBI Nucleotide BLAST was done on all the sequences to confirm the species' identity. The phylogenetic analysis and the tree were constructed by combining sequences of ITS (550 bp) and β-tubulin (1025 bp) using MEGA version 11 with the maximum likelihood method. The ex-type strain of *T. asperellum* CBS 433.97 (NR130668, XM024905825) as a reference sequence, *Chaetomium globosum*CBS 160.62 (NR144851, KT214742) as an outgroup, and other closely related species includes *T. pubescens* (KF294849, KF609264), *T. harzianum* (MK886859, MK895942), *T. hamatum* (LT707583, LT707606), *T. longibrachiatum* (JN039058, KF595268), *T. virens* (HM046563, KF595234) were used to construct the tree. To assess the stability of branches, a bootstrap analysis with 1000 replications was performed.

## 2.4. In Vitro Antagonistic Assay

The twenty *T. asperellum* isolates were tested for their in vitro antagonistic activity against *A. rolfsii* using a dual culture assay. All the *T. asperellum* isolates, and *A. rolfsii* isolate Sr38 were grown on PDA plates for 7 days at 28 ± 2 °C. On the PDA plate, a 5 mm disc of *T. asperellum* and the test pathogen (*A. rolfsii*) were placed immediately opposite one another, 2 cm from the edge, and cultured for 7 days at 28 ± 2 °C. As a control, separate petri plates containing only the pathogen (*A. rolfsii*) were kept. Three replications of each treatment were used for the whole experiment. Following Garcia's formula [36], which measures Inhibition (%) = 100 [(C − T)/C], the radial growth (mm) and percent inhibition were measured and calculated. C represents the growth of the pathogen in the control plate,

and T represents the growth of the pathogen in the dual culture plate (treatment) [37]. The impact of antibiotics generated by antagonistic fungi (*Trichoderma*) was investigated using a volatile assay [38,39]. A 5 mm *T. asperellum* mycelial disc was centered on the PDA plate, and another plate had the pathogen *A. rolfsii* inoculated in the same way. Each petri dish pair was sealed together with cellophane adhesive tape with the antagonist plate down and the pathogen plate on the upside and incubated at 28 ± 2 °C for 7 days. As a control, Petri dishes were used with the pathogen on the top plate and without the antagonist on the bottom plate. According to Garcia, the percentage of mycelial growth inhibition was calculated 7 days after inoculation.

*2.5. In Planta Bio-Control Assay*

An in planta study was conducted on one-month-old tomato plants (var. Pusa Ruby) by pre- and post-inoculation approach of *T. asperellum* isolates in the *A. rolfsii* infected soil to evaluate their antagonistic potential against the disease. The experiment was carried out at the Center for Protected Cultivation Technology (CPCT), IARI, New Delhi. The experiment was conducted by keeping four treatments for all the isolates, viz., control (treated with distilled water), biocontrol treated, pathogen treated, and pathogen + biocontrol-treated plants to compare the effect.

Sorghum grains were used to multiply the test pathogen *A. rolfsii*. The grains were initially soaked in water overnight and then drained off the excess water, filled 100 g sorghum grains in 500 mL Erlenmeyer flasks, plugged with non-absorbent cotton, and autoclaved at 15 psi at 121 °C for 30 min. After cooling at room temperature, the flasks were inoculated with mycelial discs of the 7-day-old culture of *A. rolfsii* under aseptic conditions and incubated at 28 ± 2 °C for 7 days until the sorghum grains were completely covered with *A. rolfsii* growth. The evaluation was conducted by mixing 10 g of *A. rolfsii* inoculums/ kg of soil to make the infected soil.

The seven-day-old pure culture of *T. asperellum* isolates was used after culturing on PDA plates upon incubation at 28 ± 2 °C. To prepare the spore suspension, 5 mL of sterile distilled water was added to each plate, and the spores were then scraped off with a sterile spatula; the spore suspensions were filtered, and the concentrations were adjusted to $10^8$ spores/ ml for all the isolates. The evaluation was carried out through pre- and post-inoculation approaches by applying 100 mL of *T. asperellum* spore suspension per pot as a drench and seedling root dip.

In the case of the post-inoculation approach, each *T. asperellum* isolate's spore suspension was then added separately for each isolate evaluation after 5 days of transplanting plants in the infected soil in the pot. Dipping the seedlings in *T. asperellum* spore suspension for 60 min and then transplanting the plants in the soil, which is already drenched with *T. asperellum* spore suspensions in the infected soil in case of the pre-inoculation approach.

After 10 days, the disease severity index was calculated according to the scale, which includes 0 = no disease symptoms; 1 = disease symptoms without visible fungal outgrowth; 2 = disease symptoms with visible fungal growth; 3 = partial wilting of the plant; and 4 = complete wilting and plant death. The experiment was conducted thrice, with three replications of each treatment, using a Completely Randomized Design (CRD). The data collected from the pathogen + biocontrol treatment from all the isolate's evaluation and the disease severity index (DSI) was calculated for both pre- and post-inoculation conditions. The disease severity index was calculated using the following formula,

$$\text{DSI (\%)} = \frac{\Sigma(\text{Severity score} \times \text{Number of infected plants having the sames core})}{\text{Total number of plants observed} \times \text{Maximum rating scale number}} \times 100$$

*2.6. Biochemical Analysis*

The specific activity of the hydrolytic enzymes (cellulase, β-1,3 glucanase, β-1,4 glucanase, protease, and chitinase) in association with *A. rolfsii* was investigated for all the isolates of *T. asperellum*. The enzyme assay was conducted in a minimal synthetic medium

supplemented with colloidal chitin at 0.05% (*w/v*) as sole carbon source by inoculation with 1 mL spore suspension of *T. asperellum* isolates ($2 \times 10^8$ cfu/mL) and test pathogen (*A. rolfsii*) in a 50 mL medium and incubated in a BOD cum rotary shaker at 150 rpm at $28 \pm 2\,°C$ for 7 days. The mycelial mat was separated to obtain the culture filtrate by running it through Whatman no. 42 filter paper before centrifuging at 6000 rpm for 10 min at $4\,°C$. The supernatant was immediately tested for enzyme activity after being collected into a sterilized conical flask. Enzyme activity was expressed in specific activity as IU/mg protein. The approach utilized by Bradford [40] was used to calculate the protein content of each treatment's culture supernatants. Non-enzymatic controls were performed using boiled enzymes and were subtracted from the enzymatic values. The details of substrates and standards used are given in Table 1. The amount of enzyme required to create one micromole of reducing sugar per minute per milliliter of culture supernatants was the definition of unit activity.

**Table 1.** Details of the selected enzymes used in this study.

| Enzymes | Substrates | Supplier and Product Number | Standards (1 mg/mL Stock) | OD (in nm) |
|---------|-----------|-----------------------------|---------------------------|-----------|
| Cellulase | Cellulose (0.5%) | Sigma-Aldrich—435236 | Glucose | 530 |
| β-1,3 glucanase | Laminarin (3.2 mg/mL distilled water) | Sigma-Aldrich—L9634 | Glucose | 530 |
| β-1,4 glucanase | Carboxy methyl cellulose (1%) | Himedia—GRM329 | Glucose | 575 |
| Chitinase | Colloidal chitin (0.5%) | Himedia—GRM1356 | NAG | 420 |
| Protease | Casein (1%) | Sigma-Aldrich—C9801 | Tyrosine | 280 |

### 2.6.1. Cellulase Assay

A total of 1 mL of 0.5% cellulose was suspended in 50 mM (0.05 M) citrate phosphate buffer (pH 4.8), and 1 mL of culture filtrates from various *T. asperellum* isolates were used separately in the assay mixture taken in 15 mL test tubes. At $50\,°C$, the reaction mixture was incubated for 30 min. A 3 mL solution of 1% DNS (Dinitrosalicylate) reagent was added to stop the reaction. It was then heated for 10 min at $100\,°C$ to produce the reddish-brown coloration. In place of culture filtrate, distilled water was used to create the blanks. The amount of reducing sugar released was calculated using the standard glucose curve, and the absorbance was measured at 530 nm [41].

### 2.6.2. β-1,3 Glucanase Assay

The assay mixture contains 0.5 mL laminarin (3.2 mg/mL distilled water), 1 mL of 0.05 M citrate buffer, and 0.5 mL of culture filtrates of various *T. asperellum* isolates separately. At $40\,°C$, the reaction mixture was incubated for 60 min. The reaction was stopped by mixing 2 mL of DNS and incubating in a boiling water bath for 15 min. The absorbance was measured at 530 nm and calculated using the standard glucose curve [42].

### 2.6.3. β-1,4 Glucanase Assay

A mixture of 1.0 mL of culture filtrate, 2.0 mL of 0.05 M citrate buffer (pH 4.8), and 1 mL of 1.0% carboxy methyl cellulose was combined and incubated at $55\,°C$ for 30 min in a water bath with periodic shaking. By boiling the mixture and adding 4.0 mL of the dinitro salicylic acid reagent to terminate the reaction, the absorbance was measured at 575 nm, and the enzyme activity was calculated using a standard glucose curve [43].

### 2.6.4. Chitinase Assay
Preparation of Colloidal Chitin

The colloidal chitin was made from crude chitin from shrimps using the process described by Vessey and Pegg [44]:

A total of 30 g of crude chitin (Himedia-GRM1356) were ground and washed in 500 mL of distilled water. After decanting the water, the residue was combined with 505 mL of acidified ethanol- ether combination (ethanol: diethyl ether: Hcl = 250:250:5 mL). The mixture was centrifuged for 15 min at 5 °C at 12,000 rpm, and the residue was recovered. At 0 °C, 10 mL acetone was added, followed by concentrated HCl until the residues were completely dissolved. The mixture was centrifuged for 15 min at 5 °C at 12,000 rpm, and the supernatant was collected. 1.5 L distilled water and ice was added to the supernatant, and chitin was allowed to precipitate for 2 h at 0 °C. Chitin was washed three times with 1 L of distilled water, and each time it was centrifuged. The last residue was collected, and this colloidal form of chitin was stored in an airtight container at 0 °C for subsequent use.

Enzyme Assay

A total of 1.0 mL of Mcllvaine's buffer (pH 4.0), 0.5 mL of culture filtrate (an enzyme source), and 0.5 mL of colloidal chitin suspension were added to the reaction mixture. This mixture was properly mixed before being incubated at 37 °C for 20 min in a water bath with occasional shaking. By boiling the mixture for three minutes in a water bath, the process was stopped. 3.0 mL of potassium ferric cyanide reagent was added and warmed in a boiling water bath for 15 min. The absorbance of the reaction mixture at 420 nm was used to calculate the amount of N-acetyl glucosamine (NAG) released using the standard NAG curve [45].

2.6.5. Protease Assay

The substrate (1% casein in 50 mM phosphate buffer, pH 7.0) was denatured in a water bath at 100 °C for 15 min before cooling to room temperature. Later a reaction mixture consisting of 1 mL of the substrate and 1 mL of enzyme solution was incubated for 20 min at 37 °C. To that reaction mixture, 3 mL of tri-chloro acetic acid (TCA) was added to settle down. The tubes were left at 4 °C for an hour to allow the precipitation of undigested protein. At 280 nm, the absorbance of liberated tyrosine in the filtrate was measured.

*2.7. Secondary Metabolites Profiling*

2.7.1. Extraction and Separation of Antifungal Metabolites

Seven *T. asperellum* isolates (A8, A10, A11, A12, A15, A17, and A20) were chosen according to their efficiency for the secondary metabolites profiling. They were then cultured in 100 mL of potato dextrose broth (PDB). Following inoculation, the medium was incubated for 15 days at 28 ± 2 °C while being continuously shaken at 160 rpm in an incubator shaker. The mycelial biomass was then separated by passing it through a Whatman no. 1 filter, and the filtrates were then successively extracted three times using a separating funnel with an equal volume of ethyl acetate. The extracted fractions were evaporated to produce a concentrate of ethyl acetate using a rotary evaporator (IKA® RV 10; Staufen, Germany). The remaining polar filtrates were passed through anhydrous sodium sulfate (10 g) to remove the excess water and stored for further analysis. The ethyl acetate concentrate was diluted to prepare a 5 μg/mL solution in gas chromatography-mass spectrometry (GCMS) grade ethyl acetate and subjected to analysis for volatile organic compounds.

2.7.2. GC-MS Analysis

Volatile organic compounds produced by each of the seven tested isolates were analyzed on 6850C Agilent GC-MS (Agilent Technologies®, Santa Clara, CA, USA). Following their separation by an Agilent HP-5MS column (30 m, 0.25 mm, film thickness 0.25 μm), the components was identified by the mass spectrometer based on their retention index and molecular weights. Helium gas (>99.99% purity) was utilized as the carrier gas at a flow rate of 1 mL/min and a pressure of 10 psi. Each sample (1 μL) was injected using a built-in auto-injector with a 20:1 split ratio into the gas chromatograph (GC). A GC-MS temperature program was developed that started at 40 °C, increased by 3 °C each minute to 130 °C,

and then held that temperature for 2 min. In addition, the temperature increased by 5 °C every minute until it reached 210 °C and stayed there for two minutes. The temperature was then increased by 10 °C/min to 350 °C. The total run time for each sample was 64 min. The following settings were used for the MS acquisition parameters: a solvent delay of 2 min, an E.M. voltage of 1214 V, an ion source temperature of 200 °C, electron ionization energy of 70 eV, a transfer line temperature of 200 °C, and full scan mode (50–550 amu). Identification of the compounds from their respective spectrum was done using the NIST library, matching with their accurate mass and retention index.

### 2.8. Thin Layer Chromatography (TLC) Assay

The TLC assay was conducted to identify the number of spots present in the ethyl acetate extract, which could give preliminary knowledge about the major components present in the isolates. For primary component partition, thin layer chromatography was done on Merck TLC Silica gel60 F$_{254}$ plates (Product number: 1.05554.0007). Seven metabolite extracts of *T. asperellum* isolates were spotted using capillary tubes at a height of one inch above the plate's bottom. Two different solvent systems, viz., Hexane: Ethyl Acetate (9:1) and Hexane: Ethyl Acetate (1:1) was used for the analysis to determine the best mobile phase to separate the components. Various spots were detected in the TLC plate, and the number of spots was identified using visualizing agents like iodine and UV light at 350 nm [46].

### 2.9. Data Analysis

The statistical program WASP 1.0 (Web Agri Stat Package) (https://ccari.icar.gov.in/waspnew.html) (accessed on 15 June 2023) was used to analyze the data using ANOVA to determine whether there were any differences in parameter values. Three replications of each treatment were used in the experiments. At a 5% level of significance, changes between treatments were assessed using Duncan's multiple-range test. The SRplot (Science and Research Online Plot) (https://www.bioinformatics.com.cn/en (accessed on 15 June 2023)) was used to create the heat maps by statistically analyzing the GC-MS data. The NTSYSpc-2.20e version was used to create the dendrogram by analyzing the TLC banding patterns from the metabolite profiles of seven *T. asperellum* isolates.

## 3. Results

### 3.1. Morphological Identification of T. asperellum Isolates

On the PDA medium, by incubating at 28 ± 2 °C, all the isolates grew faster, and the mycelial colonies were initially white three days after incubation (DAI) and turned greenish after sporulation on 7DAI (Table 2, Figure S1). The microscopic observation resulted in branched, mostly paired conidiophores, ampuliform phialides, and slightly ovoidal to globose light green conidia (Figure 1).

**Table 2.** Collection, identification, and NCBI GenBank accession numbers of the twenty *T. asperellum* isolates.

| Isolates Code | Collection Source and Location | GPS Location | | ITS [a] | β-Tubulin [a] | Colony Characteristics |
| --- | --- | --- | --- | --- | --- | --- |
| | | Latitude | Longitude | | | |
| A1 | Soil, IARI | 28°38′ N | 77°10′ E | OR133614 | OR193716 | Abundant mycelium with dark green spores |
| A2 | Soil, IARI | 28°38′ N | 77°10′ E | OQ892293 | OR193717 | Abundant mycelium with yellowish-green spores |
| A3 | Rhizosphere, Guntur | 16°18′ N | 80°27′ E | OR133699 | OR193718 | Less aerial mycelium with dark green centered spores |

**Table 2.** *Cont.*

| Isolates Code | Collection Source and Location | GPS Location | | ITS [a] | β-Tubulin [a] | Colony Characteristics |
|---|---|---|---|---|---|---|
| | | Latitude | Longitude | | | |
| A4 | Groundnut field, TN | 11°39′ N | 78°12′ E | OR133720 | OR193707 | Abundant mycelium with yellowish-green spores |
| A5 | Vegetable field, TN | 11°3′ N | 77°17′ E | OR133722 | OR193719 | Dense dark green spores with less aerial mycelium form rings |
| A6 | Soil, IIHR, KA | 13°7′ N | 72°29′ E | OR133723 | OR193720 | Dense dark green spores with cottony mycelium |
| A7 | Soil, Akola | 20°42′ N | 76°59′ E | OR133724 | OR193721 | Cottony mycelium with light green spores |
| A8 | Soil, Tirupathi | 13°37′ N | 79°25′ E | OR133725 | OR193708 | Dark green spores forming rings with cottony mycelium |
| A9 | Soil, Navsari | 20°57′ N | 72°55′ E | OR133728 | OR193709 | Dense light to dark greenish spores with aerial mycelium |
| A10 | Soil, Navsari | 20°57′ N | 72°55′ E | OR137590 | OR193710 | Fast-growing, dense dark green spores with aerial mycelium |
| A11 | Soil, Jaipur | 26°55′ N | 75°49′ E | OR133981 | OR193711 | Fast-growing, abundant cottony mycelium forms spores at later |
| A12 | Veg. field, IARI | 28°38′ N | 77°10′ E | OR134019 | OR193712 | Dark green-centered spores with aerial mycelium |
| A13 | Chilli field, IARI | 28°38′ N | 77°10′ E | OR134022 | OR193722 | Aerial mycelium with abundant yellowish to light green spores |
| A14 | Soil, Lucknow | 26°50′ N | 80°55′ E | OR134094 | OR193723 | Abundant aerial mycelium with dense dark green spores |
| A15 | Soil, Varanasi | 25°19′ N | 82°58′ E | OR134235 | OR193713 | Less mycelium, slow growing with very less spores |
| A16 | Soil, Barracpore | 22°45′ N | 88°22′ E | OR134238 | OR193714 | Dense light green spores with aerial mycelium |
| A17 | Soil, IIHR | 13°7′ N | 72°29′ E | OR134254 | OR193724 | Fast-growing, dense aerial mycelium forms spores at later |
| A18 | Soil, Navsari | 20°57′ N | 72°55′ E | OR134258 | OR193725 | Dense light green spores abundant throughout the plate |
| A19 | Soil, Navsari | 20°57′ N | 72°55′ E | OR134337 | OR193715 | Dense yellowish to light green spores with less mycelium |
| A20 | Soil, NBPGR | 28°38′ N | 77°10′ E | OR134338 | OR193726 | Dense dark green spores with less aerial mycelium |

[a] GenBank accession number.

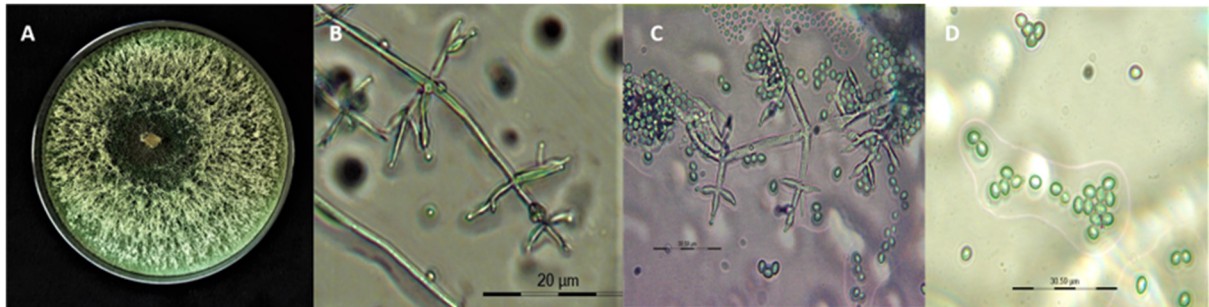

**Figure 1.** Morphological characteristics of *T. asperellum*. (**A**) Colony growth on potato dextrose agar. (**B**) Microscopic images showing conidiophores structures and branching pattern (magnification at 40×). (**C**) Phialides arrangement with conidia (magnification at 40×). (**D**) Conidia (magnification at 100×).

### 3.2. Molecular Identification and Phylogenetic Analysis

The amplified products of ITS and β-tubulin gene primers were confirmed using agarose gel electrophoresis (Figures S2 and S3) and then sequenced using the Sanger technique by outsourcing (Anuvanshiki (OPC) Pvt. Ltd.). All sequences were submitted to GenBank, and accession numbers obtained are listed in Table 2. All isolate's taxonomic identities were established by NCBI blast, and phylogenetic analysis of combined partial ITS and β-tubulin sequences was performed using the maximum likelihood method (Figure 2). The phylogenetic tree represents that the isolates A10, A8, A9, A13, A20, A7, A3, A14, A19, A1, A16, and A6 forms a clade. The isolates A5, A11, A15, A18, A4, A17, A2, and A12 form a separate clade, which shows the genetic diversity among the isolates between the above two clades. The outcome revealed that all the 20 isolates share the highest homology with *T. asperellum* Type strain CBS 433.97 followed by *T. pubescens*, *T. harzianum*, *T. hamatum*, *T. virens*, *T. longibrachiatum* with *Chetomium globosum* CBS 160.62 used as an outgroup.

### 3.3. In Vitro Antagonistic Assay

Under the dual culture assay, the mycelial growth of *A. rolfsii* was significantly inhibited by all the *T. asperellum* isolates, with the percent inhibition ranging from 20.87% to 94.66% ($p < 0.05$). Among them, the A10 isolate showed maximum inhibition of 94.66%, followed by A11 and A17, which exhibited 82.64% and 81.19% inhibition, respectively, 7 days post-inoculation. The volatile compounds produced by the *T. asperellum* isolates showed 11.55% to 70.95% mycelial inhibition. Among them, the A10 isolate showed maximum inhibition of 70.95%, followed by A17 and A11, which showed 68.27% and 63.75% inhibition, respectively (Table 3, Figures 3 and 4). The isolates A12 and A15 showed the least inhibition in both assays, which shows that they have low antagonistic potential, and the remaining isolates showed moderate mycelial growth inhibition. The comparison of the treatment means was evaluated using Duncan's multiple-range test.

**Table 3.** Effect of *T. asperellum* isolates on radial growth inhibition of *A. rolfsii* on PDA at 28 ± 2 °C at 7 days after inoculation (DAI).

| | Percent Inhibition of *A. rolfsii* Growth | | | | | | | | | |
|---|---|---|---|---|---|---|---|---|---|---|
| **Isolates** | **Dual Culture Assay** | | | | | **Volatile Assay** | | | | |
| | **R1** | **R2** | **R3** | **Average** | **SD** | **R1** | **R2** | **R3** | **Average** | **SD** |
| A1 | 64.58 | 64.22 | 64.15 | 64.32 [f] | 0.23 | 40.61 | 41.50 | 41.85 | 41.32 [k] | 0.64 |
| A2 | 64.62 | 64.50 | 63.66 | 64.26 [f] | 0.52 | 45.20 | 44.00 | 44.50 | 44.57 [j] | 0.60 |
| A3 | 66.69 | 66.20 | 65.90 | 66.26 [e] | 0.40 | 48.15 | 46.20 | 46.50 | 46.95 [i] | 1.05 |
| A4 | 68.62 | 67.50 | 68.22 | 68.11 [d] | 0.57 | 50.16 | 48.26 | 48.80 | 49.07 [h] | 0.98 |
| A5 | 68.89 | 67.82 | 66.95 | 67.89 [d] | 0.97 | 50.88 | 49.60 | 49.22 | 49.90 [h] | 0.87 |

**Table 3.** *Cont.*

| Isolates | Dual Culture Assay | | | | | Volatile Assay | | | | |
|---|---|---|---|---|---|---|---|---|---|---|
| | R1 | R2 | R3 | Average | SD | R1 | R2 | R3 | Average | SD |
| A6 | 50.61 | 51.20 | 52.33 | 51.38 [h] | 0.87 | 28.06 | 26.30 | 26.75 | 27.04 [m] | 0.91 |
| A7 | 67.10 | 67.85 | 68.45 | 67.80 [d] | 0.68 | 51.66 | 50.20 | 49.24 | 50.37 [gh] | 1.22 |
| A8 | 69.23 | 68.50 | 68.00 | 68.58 [d] | 0.62 | 53.24 | 51.76 | 51.90 | 52.30 [f] | 0.82 |
| A9 | 62.31 | 60.50 | 61.45 | 61.42 [g] | 0.91 | 46.32 | 46.90 | 46.15 | 46.46 [i] | 0.39 |
| A10 | 95.23 | 94.60 | 94.15 | 94.66 [a] | 0.54 | 72.10 | 70.15 | 70.60 | 70.95 [a] | 1.02 |
| A11 | 82.64 | 80.69 | 84.59 | 82.64 [b] | 1.95 | 65.72 | 63.16 | 62.38 | 63.75 [c] | 1.75 |
| A12 | 30.58 | 31.22 | 31.00 | 30.93 [i] | 0.33 | 22.64 | 22.55 | 21.80 | 22.33 [n] | 0.46 |
| A13 | 70.54 | 71.66 | 71.50 | 71.23 [c] | 0.61 | 60.50 | 61.25 | 61.80 | 61.18 [d] | 0.65 |
| A14 | 70.12 | 71.50 | 70.88 | 70.83 [c] | 0.69 | 60.12 | 59.00 | 59.25 | 59.46 [e] | 0.59 |
| A15 | 22.82 | 18.92 | 20.87 | 20.87 [j] | 1.95 | 11.20 | 12.30 | 11.15 | 11.55 [op] | 0.65 |
| A16 | 68.23 | 67.77 | 68.00 | 68.00 [d] | 0.23 | 59.88 | 58.20 | 56.30 | 58.13 [e] | 1.79 |
| A17 | 80.46 | 82.10 | 81.00 | 81.19 [b] | 0.84 | 69.21 | 67.24 | 68.35 | 68.27 [b] | 0.99 |
| A18 | 65.91 | 65.25 | 64.20 | 65.12 [ef] | 0.86 | 52.65 | 51.60 | 51.25 | 51.83 [fg] | 0.73 |
| A19 | 68.70 | 68.45 | 68.00 | 68.38 [d] | 0.35 | 50.12 | 49.45 | 50.00 | 49.86 [h] | 0.36 |
| A20 | 52.35 | 51.00 | 53.70 | 52.35 [h] | 1.35 | 31.22 | 29.45 | 28.60 | 29.76 [l] | 1.34 |
| CD @ 5% | | | | 1.497 | | | | | 1.603 | |
| SEm ± | | | | 0.823 | | | | | 0.943 | |
| CV (%) | | | | 1.411 | | | | | 2.034 | |

SD—Standard Deviation, CD—Critical Difference, SEm—Standard Error of mean, CV—Coefficient of variation, Different letters after values are significantly different at $p \leq 0.05$.

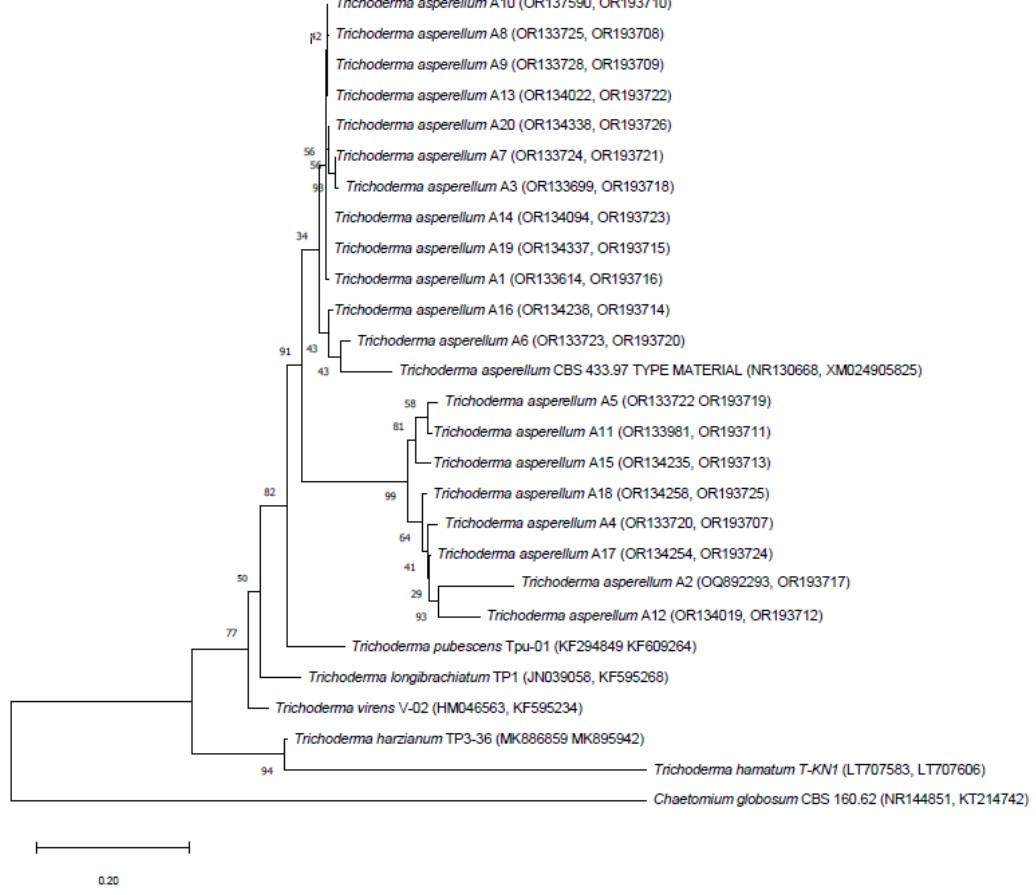

**Figure 2.** Maximum likelihood phylogenetic tree based on concatenated Internal Transcribed Spacer (ITS) and β-tubulin sequences of the 20 *T. asperellum* test isolates and its related species and reference

*T. asperellum* Type strain CBS 433.97 retrieved from the NCBI database. The tree was rooted using *Chaetomium globosum* CBS 160.62 as an outgroup.

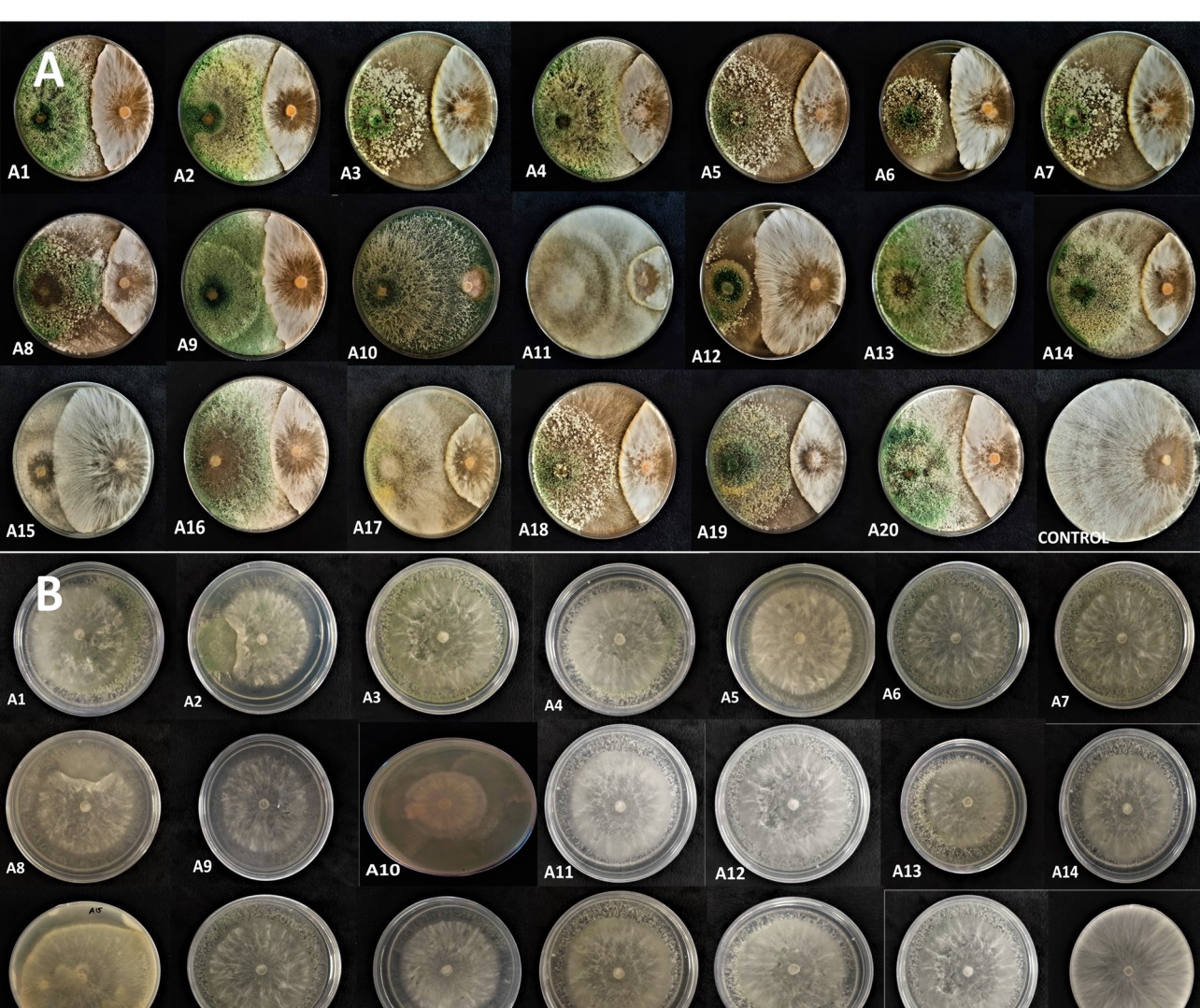

**Figure 3.** Effect of *T. asperellum* isolates on *A. rolfsii* growth through the bio efficacy tests. (**A**) Dual culture assay. (**B**) Volatile assay.

### 3.4. In Planta Bio-Control Assay

In the greenhouse pot assay, twenty *T. asperellum* isolates were evaluated against collar rot in tomatoes using a completely randomized design, and the disease severity index was recorded. The disease was significantly inhibited by all the *T. asperellum* isolates ranging from a maximum of 86.17% and 80.60 % disease inhibition from the A10 isolate to the minimum disease inhibition of 11.11% and 8.33% from the isolate A15 during pre- and post-inoculation treatment. The isolates A10 (13.83%), A11 (22.20%), and A14 (27.67%) treated plants on the *A. rolfsii* infected soil showed lower disease incidence as compared to pathogen alone treated when applied as a pre-inoculation approach against *A. rolfsii*. Pathogen treatment was maintained by inoculating only with *A. rolfsii*, which exhibited complete wilting and plant death, to compare the effect. The efficiency of *T. asperellum* isolates was also assessed by the post-inoculation approach of *T. asperellum* against *A. rolfsii*. The lowest percent of disease incidence was observed from the evaluation of A10 (19.40%), followed by A11 (27.78%) and A14 (33.33%), respectively, in post-inoculation conditions. Notably, the A15 isolate-treated plants on the infected soil recorded a maximum disease incidence

of 88.89% in pre-inoculation and 91.67% in post-inoculation. The standard deviation was recorded from each measurement to analyze the variance (Table 4, Figures 5 and 6).

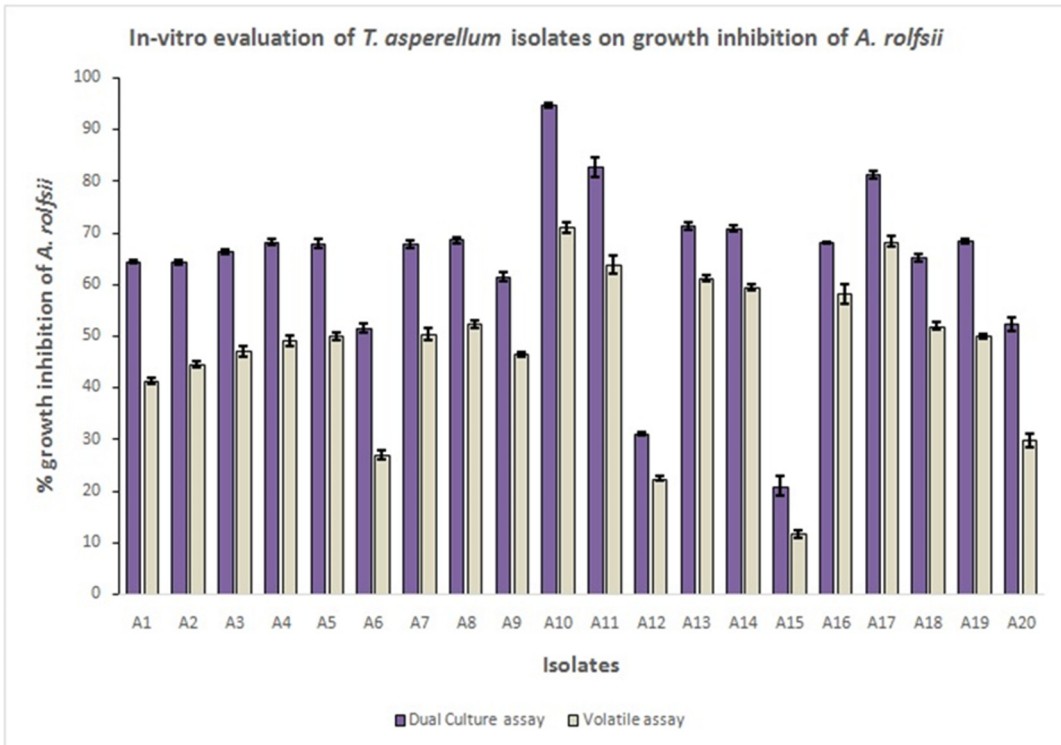

**Figure 4.** Graphical representation of the effect of *T. asperellum* isolates on the growth inhibition of *A. rolfsii*.

**Table 4.** Effect of *T. asperellum* isolates against collar rot disease in tomato plants var. pusa ruby in pot condition.

| Isolates | % Disease Severity Index (DSI) | | | | | | | | | |
|---|---|---|---|---|---|---|---|---|---|---|
| | Pre-Inoculation Treatment | | | | | Post-Inoculation Treatment | | | | |
| | R1 | R2 | R3 | Average | SD | R1 | R2 | R3 | Average | SD |
| A1 | 50 | 41.67 | 41.67 | 44.45 | 4.81 | 50 | 50 | 58.33 | 52.78 | 4.81 |
| A2 | 50 | 50 | 41.67 | 47.22 | 4.81 | 50 | 58.33 | 58.33 | 55.55 | 4.81 |
| A3 | 41.67 | 41.67 | 33.33 | 38.89 | 4.82 | 58.33 | 41.67 | 41.67 | 47.22 | 9.62 |
| A4 | 50 | 50 | 50 | 50.00 | 0.00 | 58.33 | 58.33 | 41.67 | 52.78 | 9.62 |
| A5 | 50 | 25 | 33.3 | 36.11 | 12.73 | 58.33 | 50 | 33.3 | 47.22 | 12.75 |
| A6 | 66.67 | 66.67 | 58.33 | 63.89 | 4.82 | 66.67 | 66.67 | 75 | 69.45 | 4.81 |
| A7 | 41.67 | 41.67 | 41.67 | 41.67 | 0.00 | 50 | 50 | 50 | 50.00 | 0.00 |
| A8 | 50 | 25 | 33.3 | 36.11 | 12.73 | 58.33 | 50 | 33.37 | 47.22 | 12.71 |
| A9 | 50 | 41.67 | 41.67 | 44.48 | 4.81 | 50 | 58.33 | 50 | 52.78 | 4.81 |
| A10 | 16.6 | 8.3 | 16.6 | 13.83 | 4.79 | 16.6 | 16.6 | 25 | 19.40 | 4.85 |
| A11 | 25 | 25 | 16.6 | 22.20 | 4.85 | 25 | 25 | 33.33 | 27.78 | 4.81 |
| A12 | 66.67 | 83.33 | 83.33 | 77.77 | 9.62 | 66.67 | 91.67 | 91.67 | 83.33 | 14.43 |
| A13 | 50 | 41.67 | 25 | 38.89 | 12.73 | 33.33 | 50 | 50 | 44.44 | 9.62 |
| A14 | 25 | 25 | 33 | 27.67 | 4.62 | 33.33 | 33.33 | 33.33 | 33.33 | 0.00 |
| A15 | 91.67 | 91.67 | 83.33 | 88.89 | 4.82 | 91.67 | 91.67 | 91.67 | 91.67 | 0.00 |
| A16 | 41.67 | 41.67 | 33 | 38.78 | 5.01 | 41.67 | 41.67 | 50 | 44.45 | 4.81 |
| A17 | 25 | 25 | 25 | 25.00 | 0.00 | 25 | 25 | 41.67 | 30.56 | 9.62 |
| A18 | 41.67 | 41.67 | 33.33 | 38.89 | 4.82 | 41.67 | 41.67 | 50 | 44.45 | 4.81 |
| A19 | 41.67 | 41.67 | 41.67 | 41.67 | 0.00 | 41.67 | 41.67 | 58.33 | 47.22 | 9.62 |
| A20 | 66.67 | 66.67 | 58.33 | 63.89 | 4.82 | 66.67 | 75 | 58.33 | 66.67 | 8.34 |

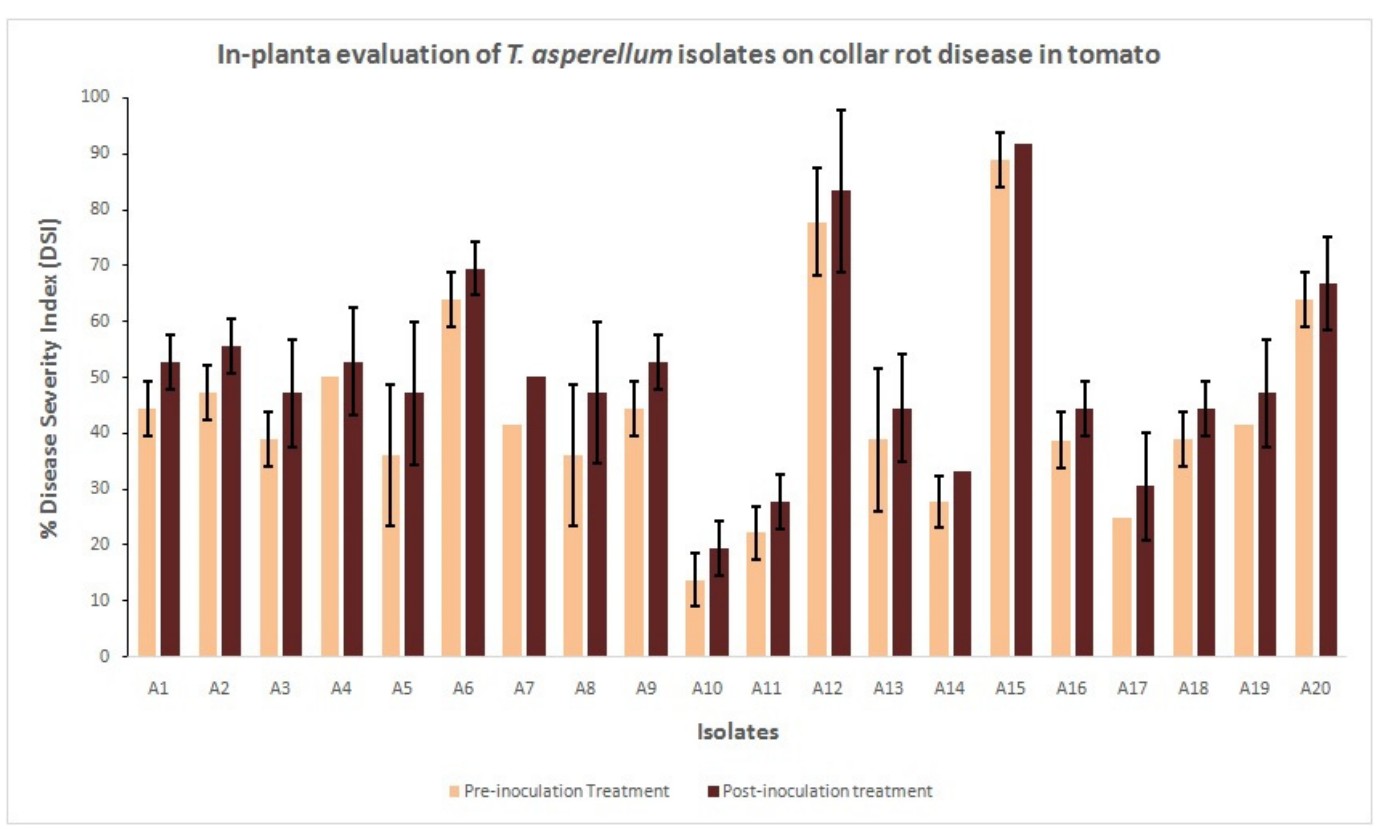

**Figure 5.** Graphical representation of the effect of *T. asperellum* isolates against collar rot disease in tomato.

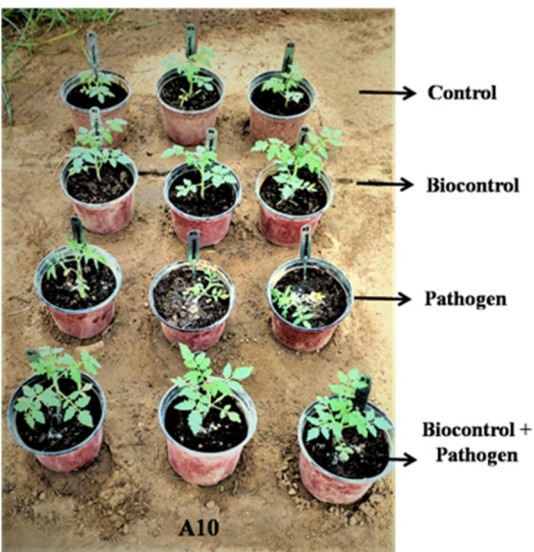

**Figure 6.** Effect of the *T. asperellum* isolate A10 on percent disease inhibition of *A. rolfsii* through the in planta evaluation.

*3.5. Enzyme Assay*

The results revealed that all the tested isolates of *T. asperellum* significantly produced different hydrolytic enzymes during their interaction with the test pathogen. The isolate A10 (174.68 IU/mg) produced the highest amount of β-1,3 glucanase, followed by A11 (149.08 IU/mg), A14 (134.68 IU/mg), and the least in A15 (24.79 IU/mg). The isolate A10 (183.48 IU/mg) showed the highest production of β-1, 4 glucanase followed by A11 (159.60 IU/mg) and A14 (147.25 IU/mg), and the least production was observed in A15

(51.28 IU/mg). The isolate A17 (80.92 IU/mg) produced the highest production of cellulase, followed by A14 (77.94 IU/mg), A16 (77.82 IU/mg), and the least production in A15 (20.84). In protease production, isolate A10 (106.06 IU/mg) showed the highest production, followed by A17 (103.08 IU/mg), and the least production was recorded in isolate A15 (15.60 IU/mg). In chitinase production, A17 (53.09 IU/mg) recorded the maximum yield, followed by A16 (52.17), and the least was observed in A15 (13.07 IU/mg) (Table 5, Figure 7).

**Table 5.** Specific activity of hydrolytic enzymes produced by *T. asperellum* isolates during the interaction with *A. rolfsii*.

| Isolates | Specific Activity in IU/mg | | | | | | | | | |
| | β-1,3 Glucanase | | β-1,4 Glucanase | | Cellulase | | Protease | | Chitinase | |
| | Avg | SD | Avg | SD | Avg | SD | Avg | SD | Avg | SD |
|---|---|---|---|---|---|---|---|---|---|---|
| A1 | 78.25 [i] | 2.00 | 79.60 [k] | 2.30 | 73.00 [cde] | 2.30 | 71.53 [f] | 2.80 | 45.07 [bcd] | 1.28 |
| A2 | 90.25 [h] | 2.00 | 79.93 [k] | 2.40 | 70.94 [ef] | 2.70 | 74.53 [ef] | 1.60 | 37.33 [fg] | 2.32 |
| A3 | 111.00 [ef] | 0.25 | 123.13 [g] | 1.80 | 75.44 [bcd] | 2.50 | 55.81 [h] | 0.80 | 7.02 [j] | 0.77 |
| A4 | 112.75 [e] | 0.25 | 109.56 [h] | 2.90 | 68.09 [fg] | 1.80 | 77.83 [d] | 2.40 | 45.85 [bc] | 2.96 |
| A5 | 125.50 [d] | 2.00 | 124.18 [fg] | 1.70 | 62.47 [h] | 3.00 | 65.33 [g] | 2.40 | 45.17 [bcd] | 2.60 |
| A6 | 59.50 [j] | 2.00 | 52.10 [m] | 2.20 | 52.04 [i] | 1.90 | 44.85 [i] | 0.80 | 39.88 [ef] | 2.13 |
| A7 | 111.58 [ef] | 1.50 | 124.28 [fg] | 2.70 | 61.92 [h] | 2.30 | 63.38 [g] | 1.70 | 36.07 [g] | 1.62 |
| A8 | 122.92 [d] | 1.80 | 127.68 [ef] | 2.50 | 76.04 [bc] | 2.10 | 66.08 [g] | 0.90 | 34.85 [gh] | 1.24 |
| A9 | 62.40 [j] | 1.70 | 88.25 [j] | 2.10 | 70.87 [efg] | 2.40 | 74.53 [ef] | 2.50 | 44.37 [bcd] | 2.52 |
| A10 | 174.68 [a] | 2.30 | 183.48 [a] | 2.40 | 77.09 [b] | 1.80 | 106.06 [a] | 1.90 | 41.87 [de] | 2.54 |
| A11 | 149.08 [b] | 2.40 | 159.60 [b] | 1.30 | 67.26 [g] | 0.70 | 82.33 [bc] | 2.40 | 46.82 [b] | 2.39 |
| A12 | 37.93 [l] | 2.80 | 30.73 [n] | 0.90 | 34.99 [j] | 1.60 | 53.53 [h] | 0.90 | 37.65 [fg] | 1.76 |
| A13 | 108.96 [fg] | 1.80 | 129.58 [e] | 1.40 | 76.75 [b] | 2.90 | 85.13 [b] | 2.40 | 42.40 [de] | 1.77 |
| A14 | 134.68 [c] | 2.10 | 147.25 [c] | 2.60 | 77.94 [ab] | 1.50 | 42.03 [i] | 1.20 | 50.81 [a] | 2.32 |
| A15 | 24.79 [m] | 2.90 | 51.28 [m] | 1.90 | 20.84 [k] | 1.90 | 15.53 [j] | 1.30 | 13.07 [i] | 1.52 |
| A16 | 107.64 [g] | 2.50 | 112.95 [h] | 1.80 | 77.82 [ab] | 2.60 | 81.35 [c] | 2.20 | 52.17 [a] | 1.95 |
| A17 | 133.18 [c] | 2.40 | 136.85 [d] | 1.60 | 80.92 [a] | 2.90 | 103.08 [a] | 1.57 | 53.09 [a] | 1.85 |
| A18 | 60.91 [j] | 1.60 | 133.43 [d] | 2.40 | 72.04 [de] | 2.50 | 75.43 [de] | 2.41 | 42.12 [de] | 1.88 |
| A19 | 112.36 [e] | 1.90 | 102.83 [i] | 1.80 | 61.75 [h] | 1.80 | 82.05 [c] | 1.23 | 31.77 [h] | 1.98 |
| A20 | 43.10 [k] | 1.80 | 58.38 [l] | 2.90 | 72.02 [de] | 1.60 | 77.15 [de] | 1.26 | 43.22 [cd] | 1.33 |
| CD @ 5% | 3.318 | | 3.543 | | 3.651 | | 3.052 | | 3.315 | |
| SEm ± | 4.044 | | 4.609 | | 4.896 | | 3.421 | | 4.035 | |
| CV (%) | 2.050 | | 1.992 | | 3.327 | | 2.647 | | 5.082 | |

Avg—Average value of three replications, SD—Standard Deviation, CD—Critical difference, SEm—Standard error of the mean, CV—Coefficient of variation, Different letters after values are significantly different at $p \leq 0.05$.

### 3.6. Comparative Analysis of Volatile Organic Compounds of T. asperellum Isolates

In the present investigation, it was revealed that the culture filtrates of the seven isolates of *T. asperellum* (A8, A10, A11, A12, A15, A17 and A20) showed the presence of many secondary metabolites at different retention times (Rt). The anti-microbial compounds were found highest from the isolate A10, 2H-pyran-2-one (17.39%) was found to be most abundant in it, followed by dienolactone (8.43%), α-pyrone (2.19%), and harziandione (0.24%) with the respective Rt of 33.48, 33.85, 33.39, and 64.23 min. 2H-Pyran-2-one (9.87%) was observed to be the most abundant in isolate A11, followed by 2,3-butanediol (7.24%) and harziandione (6.70%) with Rt of 18.65, 42.67 and 63.77 min, respectively. The isolate A17 showed many metabolites, like hexadecanoic acid (9.42%), octadecanoic acid (3.87%), hexadecane (3.83%), docosene (4.07%), and (E)-6-pen-1-enylpyran-2-one (2.18%) with the Rt of 13.67, 23.90, 19.55, 32.55 and 15.35 min respectively. 2,3-butanediol (23.41%), (E)-6-Pent-1-enyl pyran-2-one (5.88%), and 2H-pyran-2-one (2.49%) with the corresponding Rt of 64.5, 13.8 and 15.7 min respectively, were recorded to be the major compounds, followed by long-chain hydrocarbons from the isolate A8. Few anti-microbial compounds, like dimethyl disulfoxide were also observed from the isolate A12 (0.79%) and A20 (0.30%)

with the respective Rt of 33.70 and 24.45 min. No major anti-microbial compounds were detected from isolate A15, suggesting that this isolate does not produce many bioactive volatile metabolites.

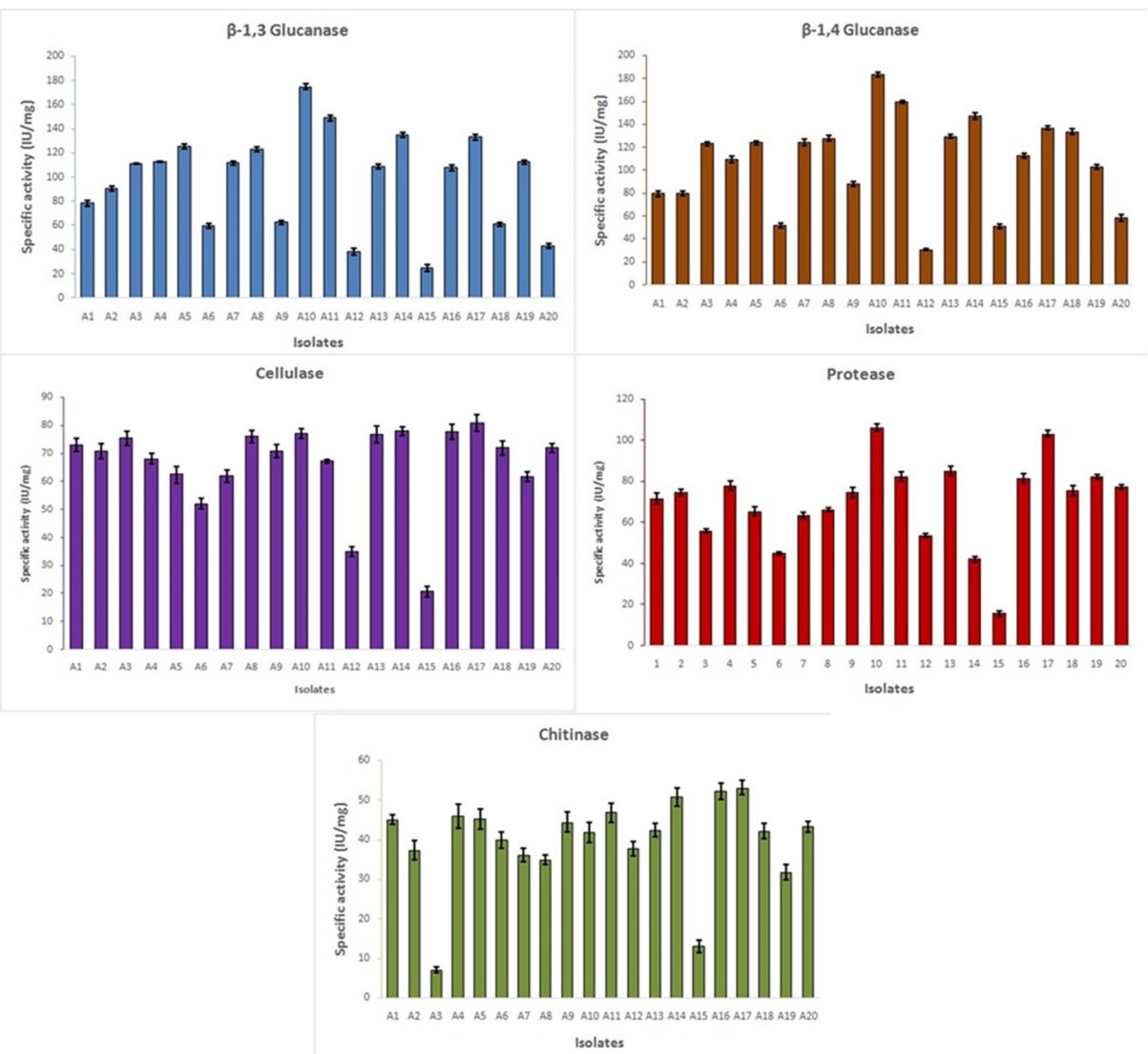

**Figure 7.** Graphical representation of the specific activity of hydrolytic enzymes produced by *T. asperellum* isolates during the interaction with *A. rolfsii*.

From the ethyl acetate fractions of *T. asperellum* isolates, the predominant compounds were identified and grouped into various functional groups with varying content ranges among isolates, including alcohols (23.68%) from the isolate A8, ketones (20.78%) from isolate A20, and acids (18.23%) from the isolate A17. Among the alcohols, 2, 3-butanediol (23.41%) was observed to be most abundant from the isolate A8. Among the Ketones having anti-microbial compounds, 2H-pyran-2-one (17.39%) was observed as the most abundant from the isolate A10, (E)-6-pent-1-enylpyran-2-one (5.88%) from the isolate A8, and α-pyrone (5.14) from the isolate A20. Among the acids, the highest content of hexadecanoic acid (9.42%) was found in the isolate A17, followed by acetic acid (3.99%) in A10, octadecanoic acid (3.87%) in A17, and 9, 12-octadecadienoic acid (2.83%) in A10. Among the lactones, δ-2,4-dienolactone (8.43%) was recorded at the maximum from isolate A10.

We observed more hydrocarbons viz., dodecane, 1-tetradecene, tetradecane, 1-hexadecene, hexadecane, cyclohexadecane, tetracosane, hexacosane, heptadecane, 1-octadecene, octadecane, docosene, tetracosahexaene from various isolates. Among them,

hexadecane (3.83%) and docosene (4.07%) were observed at the highest content in the isolate A17, whereas diterpenes, namely, harziandione (6.70%), were abundant in the isolate A11. The list of identified compounds in the ethyl acetate fraction from various *T. asperellum* isolates is listed in Table 6.

The total ion chromatograms for all the isolates are shown in Figures S4–S10. Furthermore, a heat map was generated to compare the visualization and interpretation of changes in the volatile organic compounds (VOCs) profile among the isolates (Figure 8). The area showing the red color in the heat map indicates the presence of that particular volatile organic compound having the maximum area-wise among the isolates. The most important anti-microbial compounds like 2H-pyran-2-one are abundant among the isolates A11, A12, A10, and A20. The 2,3-butanediol presence is more in the isolates A11, A12, and A8. The isolate A17 has high hexadecanoic acid. The presence of harziandione is more in the isolates A11 and A20. The isolates A20 and A10 contain high α-pyrone and 2,4 dienolactone, respectively.

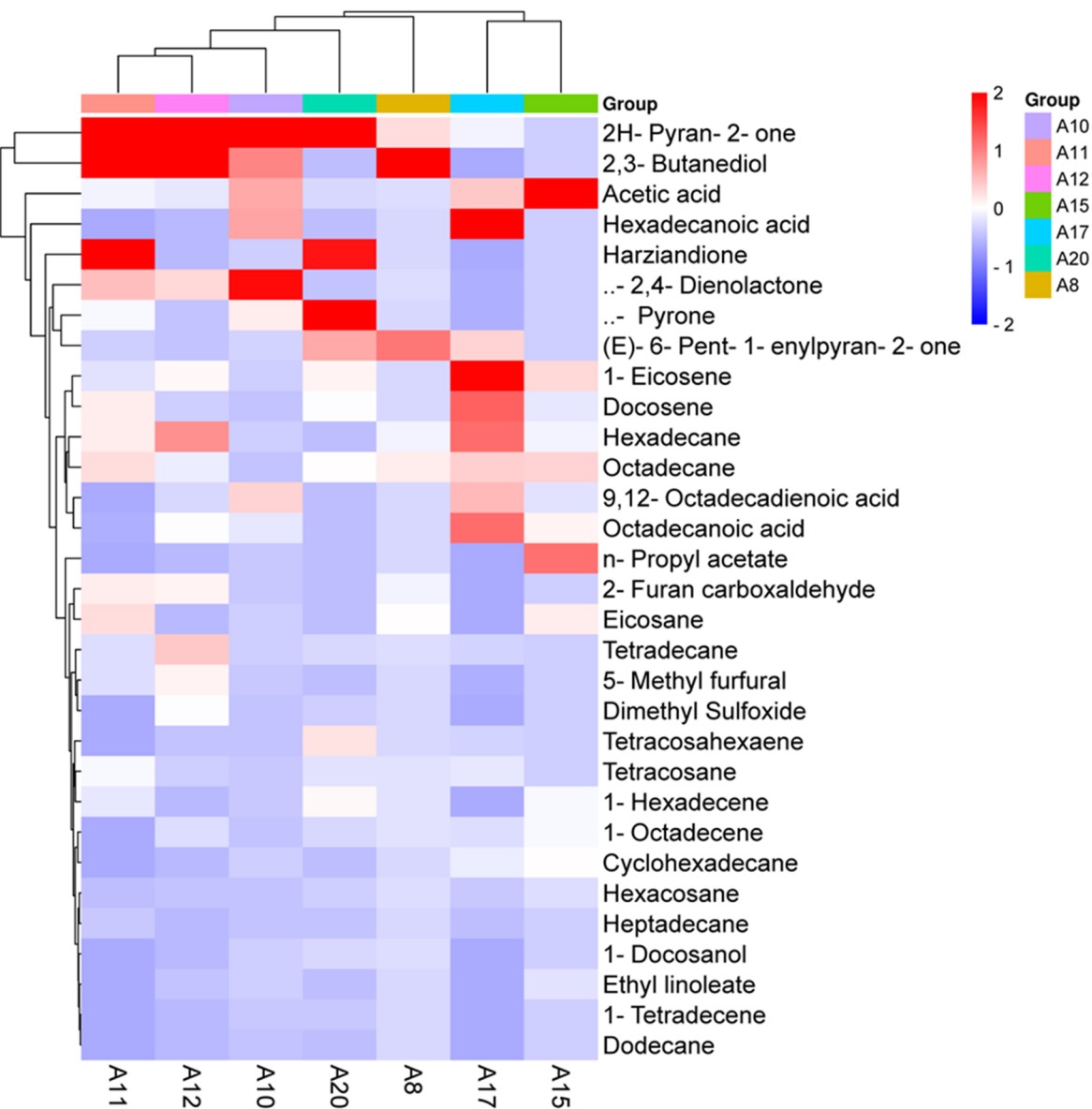

**Figure 8.** Heat map clustering of Volatile Organic Compounds (VOCs) profiles from *T. asperellum* isolates.

**Table 6.** Characteristic features of secondary metabolites isolated from *T. asperellum* isolates by GC-MS.

| Compounds | Functional Group | Molecular Formula | A10 | | A17 | | A12 | | A15 | | A8 | | A11 | | A20 | |
|---|---|---|---|---|---|---|---|---|---|---|---|---|---|---|---|---|
| | | | * RA % | RT ** | * RA % | RT ** | * RA % | RT ** | *RA % | RT ** | * RA % | RT ** | * RA % | RT ** | * RA % | RT ** |
| Acetic acid | Acid | $CH_3COOH$ | 3.99 ± 1.12 | 5.68 | 2.35 ± 0.42 | 4.83 | 0.53 ± 0.03 | 6.88 | 1.47 ± 0.39 | 12.53 | 0.15 ± 0.02 | 9.43 | 1.28 ± 0.62 | 16.52 | 0.40 ± 0.02 | 8.24 |
| *n*-Propyl acetate | Ester | $C_5H_{10}O_2$ | 0.21 ± 0.06 | 5.82 | - | - | - | - | 0.40 ± 0.02 | 13.05 | - | - | - | - | - | - |
| 2,3-Butanediol | Alcohol | $C_4H_{10}O_2$ | 4.99 ± 1.07 | 8.27 | - | - | 6.57 ± 1.32 | | - | - | 23.41 ± 4.56 | 64.5 | 7.24 ± 1.64 | 42.67 | - | - |
| 2-Furan carboxaldehyde | Aldehyde | $C_5H_4O_2$ | 0.12 ± 0.03 | 10.55 | - | - | 0.97 ± 0.05 | | - | | 0.90 ± 0.02 | 14.2 | 1.88 ± 0.51 | 24.52 | - | |
| 5-Methyl furfural | Aldehyde | $C_6H_6O_2$ | 0.12 ± 0.01 | 10.55 | 0.07 ± 0.02 | 12.65 | 0.97 ± 0.06 | | - | - | - | - | 0.94 ± 0.11 | 18.97 | - | - |
| Dodecane | Hydrocarbon | $C_{12}H_{26}$ | 0.04 ± 0.01 | 18.92 | - | - | - | - | - | - | - | - | - | - | - | - |
| 1-Tetradecene | Hydrocarbon | $C_{14}H_{28}$ | 0.12 ± 0.04 | 27.33 | - | - | - | - | - | - | - | - | - | - | 0.18 ± 0.01 | 10.32 |
| Tetradecane | Hydrocarbon | $C_{14}H_{30}$ | 0.27 ± 0.13 | 27.65 | 0.68 ± 0.14 | 26.45 | 1.41 ± 0.42 | 12.35 | - | - | 0.31 ± 0.01 | 18.9 | 0.89 ± 0.07 | 10.58 | 0.44 ± 0.02 | 11.47 |
| α- Pyrone | Ketone | $C_5H_4O_2$ | 2.19 ± 0.82 | 33.39 | 0.11 ± 0.03 | 12.88 | 0.12 ± 0.01 | 18.43 | - | - | 0.04 ± 0.01 | 12.7 | 1.43 ± 0.22 | 20.34 | 5.14 ± 1.94 | 22.25 |
| δ-2,4-Dienolactone | Lactone | $C_7H_{12}$ | 8.43 ± 1.53 | 33.85 | 0.06 ± 0.01 | 8.69 | 1.23 ± 0.47 | 15.30 | - | - | 0.21 ± 0.05 | 8.4 | 2.76 ± 0.13 | 12.76 | 0.13 ± 0.04 | 7.86 |
| 2H-Pyran-2-one | Ketone | $C_5H_4O_2$ | 17.39 ± 3.45 | 33.48 | 1.24 ± 0.32 | 13.80 | 5.23 ± 1.94 | 19.21 | - | - | 2.49 ± 0.86 | 15.7 | 9.87 ± 2.41 | 18.65 | 8.89 ± 2.32 | 20.34 |
| (E)-6-Pent-1-enylpyran-2-one | Ketone | $C_{21}H_{26}O_2$ | 0.43 ± 0.05 | 34.55 | 2.18 ± 0.48 | 15.35 | 0.1 ± 0.02 | 17.33 | - | - | 5.88 ± 1.23 | 13.8 | 0.59 ± 0.04 | 18.43 | 2.26 ± 0.91 | 13.66 |
| 1-Hexadecene | Hydrocarbon | $C_{16}H_{32}$ | 0.12 ± 0.02 | 35.49 | - | - | - | - | 0.09 ± 0.01 | 15.92 | 0.49 ± 0.03 | 27.3 | 1.09 ± 0.32 | 18.60 | 1.11 ± 0.07 | 15.97 |
| Hexadecane | Hydrocarbon | $C_{16}H_{32}$ | 0.33 ± 0.08 | 35.74 | 3.83 ± 0.71 | 19.55 | 2.07 ± 0.72 | 14.73 | 0.08 ± 0.01 | 15.95 | 0.99 ± 0.12 | 27.6 | 1.91 ± 0.62 | 19.71 | - | - |
| Cyclohexadecane | Hydrocarbon | $C_{16}H_{32}$ | 0.25 ± 0.02 | 49.10 | 1.09 ± 0.32 | 12.75 | - | - | 0.11 ± 0.03 | 15.91 | - | - | - | - | - | - |
| 1-Eicosene | Ester | $C_{20}H_{40}$ | 0.25 ± 0.04 | 49.10 | 5.63 ± 1.79 | 5.68 | 0.86 ± 0.06 | 28.45 | 0.19 ± 0.04 | 5.43 | - | - | 1.03 ± 0.04 | 29.56 | 1.18 ± 0.11 | 24.70 |
| 1-Docosanol | Alcohol | $C_{22}H_{46}O$ | 0.25 ± 0.01 | 49.10 | - | - | - | - | - | - | 0.27 ± 0.07 | 11.3 | - | - | 0.46 ± 0.06 | 26.85 |
| Eicosane | Hydrocarbon | $C_{20}H_{42}$ | 0.25 ± 0.02 | 49.28 | - | - | - | - | 0.14 ± 0.04 | 8.35 | 1.40 ± 0.09 | 8.25 | 2.10 ± 0.90 | 28.98 | - | - |
| Hexadecanoic acid | Acid | $C_{16}H_{32}O_2$ | 4.12 ± 1.23 | 50.29 | 9.42 ± 2.96 | 13.67 | - | - | - | - | - | - | - | - | - | - |
| 9,12-Octadecadienoic acid | Acid | $C_{18}H_{32}O_2$ | 2.83 ± 0.90 | 55.19 | 2.59 ± 0.41 | 24.56 | 0.37 ± 0.08 | 8.43 | 0.04 ± 0.01 | 18.90 | - | - | - | - | - | - |
| Ethyl linoleate | Ester | $C_{20}H_{36}O_2$ | 0.35 ± 0.12 | 54.41 | - | - | 0.14 ± 0.04 | 12.56 | 0.04 ± 0.01 | 13.50 | - | - | - | - | - | - |
| Tetracosane | Hydrocarbon | $C_{24}H_{50}$ | 0.11 ± 0.02 | 60.65 | 1.06 ± 0.02 | 13.80 | 0.26 ± 0.03 | 13.45 | - | - | 0.33 ± 0.04 | 6.76 | 1.37 ± 0.46 | 13.08 | 0.58 ± 0.04 | 27.59 |

**Table 6.** *Cont.*

| Compounds | Functional Group | Molecular Formula | A10 * RA % | A10 RT ** | A17 * RA % | A17 RT ** | A12 * RA % | A12 RT ** | A15 *RA % | A15 RT ** | A8 * RA % | A8 RT ** | A11 * RA % | A11 RT ** | A20 * RA % | A20 RT ** |
|---|---|---|---|---|---|---|---|---|---|---|---|---|---|---|---|---|
| | | | | | | | | | | | | | | | | |
| Harziandione | Ketone | $C_{20}H_{30}O_2$ | 0.24 ± 0.01 | 64.23 | - | - | - | - | - | - | - | - | 6.70 ± 1.94 | 63.77 | 4.49 ± 2.07 | 58.88 |
| Hexacosane | Hydrocarbon | $C_{26}H_{54}$ | 0.05 ± 0.01 | 65.71 | 0.57 ± 0.12 | 11.65 | 0.14 ± 0.01 | 19.58 | 0.03 ± 0.01 | 2.56 | 0.15 ± 0.03 | 12.34 | 0.33 ± 0.04 | 19.42 | 0.25 ± 0.03 | 15.20 |
| Heptadecane | Hydrocarbon | $C_{17}H_{36}$ | - | - | 0.32 ± 0.08 | 13.09 | - | - | - | - | - | - | 0.56 ± 0.05 | 18.91 | 0.12 ± 0.02 | 10.24 |
| 1-Octadecene | Hydrocarbon | $C_{18}H_{36}$ | - | - | 0.83 ± 0.27 | 12.60 | 0.41 ± 0.02 | 13.08 | 0.09 ± 0.02 | 14.29 | 0.44 ± 0.06 | 14.0 | - | - | 0.46 ± 0.07 | 8.65 |
| Octadecane | Hydrocarbon | $C_{18}H_{36}$ | - | - | 2.23 ± 1.01 | 12.60 | 0.62 ± 0.03 | 15.48 | 0.20 ± 0.03 | 9.20 | 1.91 ± 0.23 | 13.2 | 2.16 ± 0.91 | 16.77 | 1.07 ± 0.38 | 18.55 |
| Docosene | Hydrocarbon | $C_{22}H_{44}$ | - | - | 4.07 ± 1.53 | 32.55 | 0.24 ± 0.03 | 11.35 | 0.06 ± 0.01 | 12.11 | - | - | 1.86 ± 0.11 | 15.69 | 0.98 ± 0.23 | 13.34 |
| Octadecanoic acid | Acid | $C_{18}H_{36}O_2$ | 0.95 ± 0.23 | 55.70 | 3.87 ± 1.25 | 23.90 | 0.75 ± 0.11 | 11.87 | 0.13 ± 0.02 | 7.88 | - | - | 0.07 ± 0.01 | 7.24 | - | - |
| Tetracosahexaene | Hydrocarbon | $C_{24}H_{38}$ | - | - | 0.72 ± 0.09 | 24.76 | 0.15 ± 0.01 | 18.68 | - | - | - | - | - | - | 1.44 ± 0.71 | 14.21 |
| Dimethyl Sulfoxide | Organosulfur | $C_2H_6OS$ | - | - | - | - | 0.79 ± 0.06 | 33.70 | - | - | - | - | - | - | 0.30 ± 0.06 | 24.45 |
| Chemical groups | | | Content (%) | | | | | | | | | | | | | |
| Acids | | | 11.89 | | 18.23 | | 1.65 | | 1.64 | | 0.15 | | 1.35 | | 0.40 | |
| Esters | | | 0.81 | | 5.63 | | 1.0 | | 0.63 | | - | | 1.03 | | 1.18 | |
| Alcohols | | | 5.24 | | - | | 6.57 | | - | | 23.68 | | 7.24 | | 0.46 | |
| Aldehydes | | | 0.24 | | 0.07 | | 1.94 | | - | | 0.90 | | 2.82 | | - | |
| Hydrocarbons | | | 1.54 | | 15.4 | | 5.3 | | 0.80 | | 6.02 | | 12.27 | | 6.63 | |
| Ketones | | | 20.25 | | 3.53 | | 5.45 | | - | | 8.41 | | 18.59 | | 20.78 | |
| Lactones | | | 8.43 | | 0.06 | | 1.23 | | - | | 0.21 | | 2.76 | | 0.13 | |
| Organosulfur | | | - | | - | | 0.79 | | - | | - | | - | | 0.30 | |
| Total | | | 48.40 | | 42.92 | | 23.93 | | 3.07 | | 39.37 | | 46.06 | | 29.88 | |

* RA %—Relative Area %; ** RT—Retention time (min.) of each volatile compound eluted through HP 5MS column in GC-MS.

### 3.7. TLC Assay

We observed that there were more bands in Hexane: Ethyl Acetate (1:1) solvent system than in the 9:1 solvent system. The A10 isolate produced nine bands in a 1:1 system, followed by the isolates A17 and A11, which produced seven bands that represent the presence of major metabolites in the ethyl acetate extract. The least number of bands observed from isolate A15 was 4 (Figure S11). Using the UPGMA (Unweighted pair group method with arithmetic mean) method in NTSYSpc-2.02e, a dendrogram was performed based on the unique banding patterns from each isolate of the *T. asperellum* metabolome. Two clades were obtained viz., A17, A10, and A11 (more promising isolates) forming one, and A8, A20, A12, and A15 (moderate to less promising isolates) forming the other, as seen in the dendrogram (Figure 9).

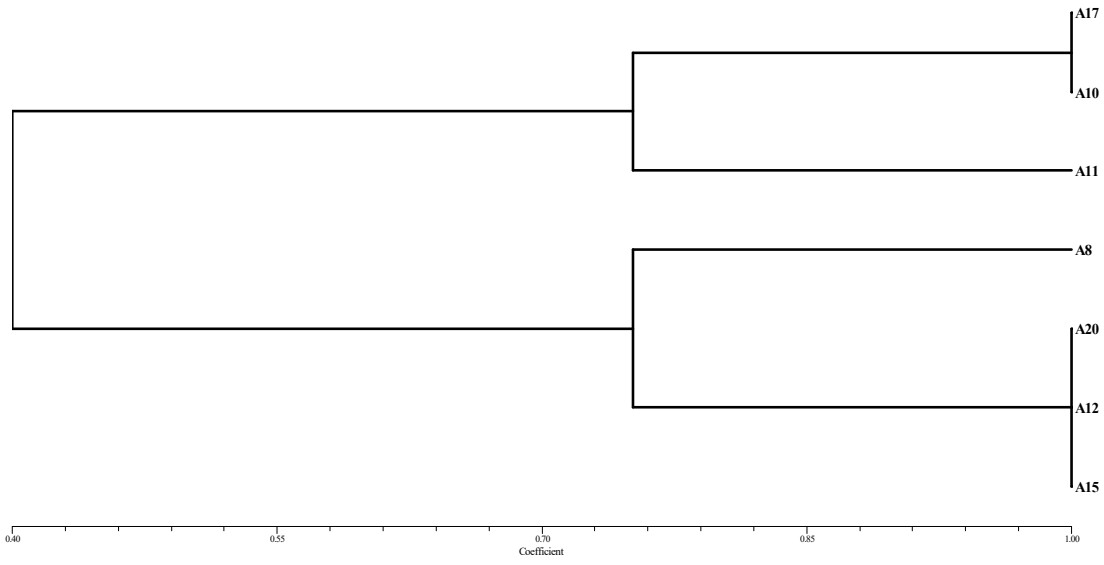

**Figure 9.** Dendrogram of seven *T. asperellum* isolates based on bioactive compounds using thin layer chromatography (Hexane: Ethyl Acetate (1:1) solvent system and long UV light range).

## 4. Discussion

In our study, 20 isolates were used as antagonists and were confirmed as *T. asperellum* based on morphological and molecular characteristics. The studied morphological characteristics, such as colony characteristics like a faster growth rate, globose conidia, mostly paired branches, and an ampulliform phialides nature, were also confirmed with the previous reports [47,48]. Conventional morphological approaches may not perform well for accurate identification. So, combining the morphological and molecular strategies is an ideal tool [49]. Using the ITS and β-tubulin markers, the precise species identification of *T. asperellum* was achieved, demonstrating the efficacy of the molecular markers in ensuring reliable species determination.

Further, these isolates were tested against the collar rot pathogen, *A. rolfsii*, indicating that *T. asperellum* is an effective biocontrol agent. In the In vitro antagonistic assay, isolate A10 recorded a maximum inhibition of *A. rolfsii* growth (94.66%) in the dual plate assay and 70.95% in the volatile assay. These results are in parallel with the previous reports, showing that most temperature-tolerant *T. asperellum* isolates showed a 50–68% reduction of hyphal growth of *A. rolfsii* under in vitro [50], and *T. asperellum* caused significant reduction in mycelial growth of 43.57%, 38.16%, and 54.87% obtained for *Pythium aphanidermatum*, *Pythium debaryanum*, and *A. rolfsii* Sr3 respectively [51]. The Tricho05 and Tricho06 isolates also recorded the maximum inhibition of *A. rolfsii* mycelial growth of 70.37% and 63.33%, respectively [52]. The In planta study carried out using tomato plant var. pusa ruby suggests that the application of *T. asperellum* to the soil prior to the development of the disease will reduce the incidence of collar rot significantly, whereas the pathogen alone treated plants showed 100% disease incidence with prominent symptoms. The isolate

A10 application on the infected soil observed lower disease incidence (13.83%) in pre-inoculation and 19.40% in post-inoculation as compared to pathogen-alone treatment. These are correlated with previous studies reports that the presence of *T. asperellum* reduced the severity of disease symptom caused by *A. rolfsii* in onion [53], and *T. asperellum* strains mixture treated cucumber plants showed the highest reduction of Fusarium root and stem rot disease incidence and severity [54].

This indicates that the biocontrol agent, *T. asperellum* inhibits the pathogen by various mechanisms, including the competing nature of space and nutrients and the production of volatile metabolites, thereby restricting the growth of the pathogen [55–57] and providing resistance against pathogens by inducing defense responses in the plants [58–61].

The production of hydrolytic enzymes to dissolve the fungal mycelial cell wall has been attributed to biocontrol activity. According to Inbar and Chet [62], the antagonistic action of *Trichoderma* species against soil-borne diseases may be related to their synthesis of these enzymes. In the present investigation, the *T. asperellum* isolates were evaluated for their potential to produce various extracellular enzymes against *A. rolfsii*. Extracellular enzyme activity was observed in all the isolates. The isolate A10 observed the maximum production of hydrolytic enzymes by the production of β-1,3 glucanase (174.68 IU/mg), β-1,4 glucanase (183.48 IU/mg), and protease (106.06 IU/mg), which correlates with the previous studies showed that *T. asperellum* isolates Ts39, Ts12, Ts42, Ts9, Ts32, Ts36 possessed the wide range of hydrolytic enzyme activities exhibiting chitinase 8.7–10.3 Pmol/s/mL and β-1,3 glucanase (1.4–1.98 nmol/s/mL) [63] and *T. asperellum* CCTCC-RW0014 reported that the production of various hydrolytic enzymes with higher enzyme activity viz., chitinase (87.5 ± 3.21%), protease (52.9 ± 1.23%), cellulase (84.8 ± 2.56%) and β-1.3 glucanase (60.5 ± 2.32%) [64] and *Trichoderma* isolates T2, T8, T9, T10, T11, T16, T17, and T20 recorded the strong production of chitinase, β-1,3 glucanase, protease, and cellulase that exhibited high inhibitory activity against *A. rolfsii* in dual culture [65]. The *T. virens* isolates V-19, and V-21 exhibited the highest enzyme activities like cellulase (30.31 IU/mg, 16.75 IU/mg), β-1,3 glucanase (19.01 IU/mg, 16.46 IU/mg), chitinase (24.21 IU/mg, 22.71 IU/mg), protease (17.13 IU/mg, 18.20 IU/mg) respectively, and *T. harzianum* isolate H10 also possessed the highest enzyme activity viz., cellulase (18.09 IU/mg), β-1,3 glucanase (16.44 IU/mg), β-1,4 glucanase (17.81 IU/mg), Chitinase (88.80 IU/mg), Protease (23.88 IU/mg) recorded during interation with *A. rolfsii* [66].

It is known that *Trichoderma* spp. produces a variety of bioactive secondary metabolites with anti-microbial properties against a variety of phytopathogens, primarily soil-borne pathogens [67]. In the present study, secondary metabolites were profiled using GC-MS analysis from seven *T. asperellum* isolates selected based on the bio-efficacy tested. 2H-Pyran-2-one, dienolactone, α-pyrone, and harziandione are the volatile secondary metabolites that have antifungal activity and are produced by *Trichoderma* spp. From our GC-MS study, we found that the maximum production of antifungal VOCs from isolate A10 is 2H-pyran-2-one, dienolactone, α-pyrone, and harziandione. Among them, 2H-Pyran-2-one is the most important compound, having antifungal activity against various plant pathogens as per the previous studies. The compound 6-pentyl-2H-pyran-2-one (6-PP), an important volatile compound recorded from the *T. asperellum* isolates Ta2, Ta12, Ta17, Ta20, and Ta45 through GC-MS analysis having the peculiar character of coconut odor with anti-microbial activity [68]. The 6-PP recorded from the *T. asperellum* possessing a powerful antifungal compound against *Magnaporthiopsis maydis* and the purified 6-PP compound (30μg/seed) used in seed coating, and the treatments reduced the symptoms (up to 20%), pathogen infection (94–98%) and improved plant biomass by 90–120% and cob weight by 60% [69]. The 6-PP has a significant inhibitory effect on turf dollar spot and recorded good control efficacy recorded both in in-vitro and in-vivo studies [70]. The 6-PP reported from *T. asperellum* PT-15 showed the antifungal effects by applying a nutrient solution containing 25 mg/L 6-PP significantly suppressed the fusarium wilt with 70.71% efficacy and a 27.23% disease index [71].

A TLC assay was conducted to identify the number of bands present in the ethyl acetate extract and to analyze the metabolites produced by the biocontrol agents under two solvent systems [72,73]. TLC results exhibited many bands from tested isolates representing major metabolites present in the ethyl acetate extract. Among the two solvent systems used for the purpose, hexane: ethyl acetate (1:1) was found to be suitable for the separation of the metabolites, which correlates with the previous report confirmed the presence of metabolites using hexane: ethyl acetate (1:1) solvent system and verified the detection of carboxylic acid as it formed as yellow spot when sprayed using bromocresol green in laccase mediated Reactive Black 5 biodegradation by the *T. atroviride* F03 [74].

This solvent system is then recommended for the isolation of the metabolites for further structural elucidation using various spectrometric techniques. However, from the TLC analysis, we observed the formation of two clusters, viz., high promising isolates as one cluster and moderate to low promising isolates as another cluster, in the dendrogram generated based on banding patterns among the isolates. We may conclude from these findings that there is a high correlation between metabolite profiles, hydrolytic enzyme production, space, nutrient competitiveness, priming of defenses in plants, and their potential as biological agents against *A. rolfsii*.

## 5. Conclusions

In recent years, biological control has received increasing attention as a promising alternative to chemical control of plant pathogens. In the present study, *T. asperellum* isolate A10 was found to be a promising antagonist for the management of *A. rolfsii* when tested under lab and greenhouse conditions. Our study also suggested that the application of *T. asperellum* before the establishment of the disease reduces the incidence of collar rot significantly. GC-MS analysis revealed the production of various antifungal secondary metabolites such as 2H-Pyran-2-one, dienolactone, α-pyrone, harziandione, butanediol, and dimethyl disulfoxide involved in the antagonistic mechanism. We found that isolate A10 produced the 2H-Pyran-2-one with the maximum relative area and retention time, playing a pivotal role in the antagonistic activity. We also observed that isolate A10 produced the maximum production of various hydrolytic enzymes like β-1,3 glucanase, β-1, 4 glucanase, and protease, which are involved in antagonistic effects on *A. rolfsii*. Our findings collectively imply that *T. asperellum* and its anti-microbial substances have great potential to successfully manage collar rot diseases. Since BCAs perform differently under varied climatic conditions, there is a need to evaluate their efficacy at multiple locations and across multiple crops to confirm their robustness and potential.

**Supplementary Materials:** The following supporting information can be downloaded at: https://www.mdpi.com/article/10.3390/horticulturae9101116/s1, Figure S1: Growth characteristics of different isolates of Trichoderma asperellum on potato dextrose agar (PDA) after 7 days of incubation at 28 $\pm$ 2 °C; Figure S2: PCR amplification patterns in ITS1-5.8S-ITS2 with ITS1/ITS4 primers from 20 Trichoderma asperellum isolates; L is 1000-bp DNA ladder; Figure S3: PCR amplification patterns in β-tubulin with B-tubf1-F / B-tubr1-R primers from 20 Trichoderma asperellum isolates; L is 1000-bp DNA ladder; Figure S4–S10: GC-MS chromatogram of secondary metabolites from T. asperellum isolates; Figure S11: TLC plates showing many spots of metabolites (A) Long UV light range (B) Short UV light range in the solvent system: Hexane: Ethyl acetate (9:1) (C) Long UV light range (D) Short UV light range in the solvent system: Hexane: Ethyl acetate (1:1).

**Author Contributions:** C.S., D.K. and R.G. (Robin Gogoi) were involved in the conceptualization of the project, study design, and critical inputs. C.S. contributed to the lab work and statistical analysis and wrote the first draft. C.S., D.K., A.K., A.D. and R.G. (Robin Gogoi) finalized the outline and prepared schematics. P.K.S., Z.H. and A.D. helped in carry out an in-vivo experiment. C.S. and A.K. carried out GC-MS work and analysis. C.S., P.R.S., M.C. and R.G. (R. Gangaraj) were helped with statistical analysis and editing of the manuscript. All authors contributed to the article and approved the submitted version. All authors have read and agreed to the published version of the manuscript.

**Funding:** This work was partially funded by the NAHEP-CAAST project on "Genomics assisted crop improvement and management". C. Shanmugaraj received Junior and Senior Research Fellowships from the Indian Council of Agricultural Research (ICAR) for pursuing the Ph.D. program at ICAR-IARI, New Delhi.

**Data Availability Statement:** The original contributions presented in the study are included in the article/Supplementary material; further inquiries can be directed to the corresponding author.

**Acknowledgments:** C. Shanmugaraj offers sincere thanks to the ICAR for financial support in the form of Junior and Senior Research Fellowships for the Ph.D. program. C. Shanmugaraj and Deeba Kamil are grateful to NAHEP-CAAST for financial assistance.

**Conflicts of Interest:** The authors declare that the research was conducted in the absence of any commercial or financial relationships that could be construed as potential conflict of interest.

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
