# Peer review of "Exploring the Potential Biocontrol Isolates of Trichoderma asperellum for Management of Collar Rot Disease in Tomato"

_horticulturae, doi:10.3390/horticulturae9101116_

Round 1

Reviewer 1 Report

The results of the manuscript are very valuable, however, the current manuscript need to be improved for publication. Herewith my main comments:

1. The authors need to use the current taxonomic name for S. rolfsii which is Agroathelia rolfsii. 

2. There is no statistical analyses for the results of this study, the data needs to be validated with statistical analyses, calculating the percentages is not is not valuable for publication. 

3. There is no re-isolation of both pathogen and Trichoderma from the In Planta assay. This is necessary to validate the outcome of the experiment.

4. There is no analytic methods and results for the phylogenetic tree. 

5. Discussion need to be revised and re-written by following below structure: 

In general each paragraph needs to follow the bellow structure:

Topic sentence for the paragraph

P1. Outcome of this study

P2. Comparing with the other studies

P3. Conclusion

Please refere to more comment in the PDF. 

Author Response

Current name Agroathelia rolfsii and taxonomic rank

Corrected and incorporated in the Manuscript

Add these to the beginning of the paragraph

Corrected and incorporated in the Manuscript

I recommend to make separate paragraph here

P1: Growth promote and its enzymes

P2: Plant defense and its enzymes

Corrected in the manuscript

Sentence never starts with abbreviation, when start the sentence, always must the genus name be written complete

Corrected in the manuscript

You mean Trichoderma??

Yes

Where this isolate was obtained from

We collected and isolated in the present study (20 isolates), the description is in Table No. 2.

This is for both Sclerotinia and Trichoderma

In the present study morphological and molecular study conducted for T. asperellum.

The test pathogen S. rolfsii is already characterized and GenBank accession number is also obtained and explained in the materials part

What software was used

Included in the manuscript

There is no information about

How many isolates

Where isolates were obtained

Accession number

20 isolates and their details are in Table No. 2

This section is very confusing and need to be rewritten (In-planta biocontrol assay)

Corrected in the manuscript (Rewritten)

RCBD (No statistical information here)

Corrected in the manuscript

Fig 1 (need to be replaced with a good quality)

Replaced

There is no explanation in analyses, here you need to explain what got in your phylogenetic tree

Included in the manuscript

Outsourcing

Included in manuscript

There is no statistical data here, were they significantly different based on p-value?

Corrected in the manuscript

Discussion part

Corrected and incorporated in the Manuscript

Reviewer 2 Report

The manuscript titled Exploring the potential biocontrol isolates of Trichoderma asperellum  for management of collar rot disease in tomato is devoted to the isolation, identification, and characterization of soil and rhizosphere Trichoderma asperellum isolates as potent biocontrol candidates of collar rot disease caused by Sclerotium rolfsii.

The manuscript provides interesting results, however, despite the large work done to get the aims of this research, some major corrections should be made throughout the manuscript to make it suitable for publication:

Comments to Authors

Introduction

Line 45-46: The sentence can be deleted

Line 55-57: Add reference

Line 62: delete “fungus species in the hypocreaceae family”

Line 61-63: Add reference

Line 64: Arabidopsis

Line 68: delete “induced”

Line 67-68: The sentence seems similar to the second part of the previous sentence: “ISR induced by cell-free culture filtrates combines the SA and JA (Jasmonic Acid)/ET (Ethylene) signaling pathways” Please correct

Line 72: Rhizoctonia solani

Line 73-77: The sentence is too long. Please rephrase.

Materials and Methods

Line 88: Rhizosphere soil of which plant(s)?

Line 90-95: The text can be deleted and rephrased as follows: “Isolation of the potential fungal biocontrol isolates from soil samples was performed on Potato Dextrose Agar (PDA) according to the procedure described by ....(add reference). Plates were incubated at 28±2°C for 5-7 days.”

Line 97: Add the number of the obtained fungal isolates. What criterion did you use to select isolates for further work?

Line 97: obtained isolates

Line 98-101: Rephrase the sentence as follows: “A highly virulent isolate S. rolfsii Sr38, obtained during our previous work (ITCC Code: 8665), was used to evaluate bicontrol potential of the obtained fungal isolates.” Add reference for S. rolfsii Sr38 if it is already published.

Line 103-107: Rephrase the text as follows: “Morphological identification of the potential biocontrol fungal isolates was performed under a compound microscope (add manufacturer). Isolates were identified based on conidiophores branching, the shape and size of the phialides, and the conidia.”

Line 105-107: The sentences can be deleted

Line 109-112: The text can be deleted or moved to Introduction or Discussion section.

Line 113-114: “They were molecularly confirmed through the amplification of the 113 ITS1/4 and β-tubulin regions.” The sentence can be deleted.

Line 114-116: Rephrase the sentence as follows: “DNA from all (add number) isolates was extracted using a CTAB (Cetyl Trimethyl Ammonium Bromide) procedure given by Cullings et al. [14].”

Line 116: by amplification

Line 121: consisted of

Line 124: delete “template was used for the amplification”

Line 135: confirm the species identity

Delete the sentence: “We proceeded with all the isolates for further evaluation.”

Line 136: Rephrase the sentence as follows: “The maximum likelihood phylogenetic tree was constructed in MEGA11 software based on the concatenated sequences of ITS and β-tubulin.”

Specify the final sequence lenghts (nt) of ITS and β-tubulin sequences used for the phylogenetic analysis.

Line 138: Include more T. asperellum strains in the phylogenetic analysis

Line 146: PDA is enough. You already explained the abbreviation.

Line 163: on one-month-old tomato plants

Line 209-216: If the assay was performed according to the completely same procedure as described in reference No. 43, there is no need to write the complete procedure. If there are some modifications from the original (cited) protocol, please specify. The same comments stands for subsections 2.6.2.-2.6.4.

Line 247: analyze?

Line 248-250: cultured in PDB. Delete “500 ml Erlenmeyer flask that contained” and “which had 249 previously been autoclaved at 121°C for 40 min”

Explain the abbreviation PDB upon the first use.

Line 258: Explain the abbreviation GCMS upon the first use.

Lines 261-262: Rephrase the sentence as follows: “Volatile organic compounds produced by each of the seven tested isolates were analysed on 6850C Agilent GC-MS (Agilent Technologies®, USA).”

Line 278-279: The sentence can be deleted.

Line 277: Thin Layer Chromatography assay (TLC)

Results

Line 301: All the isolates grew at a faster rate on the PDA medium“ – explain

Line 300-305: When reffering to the Table 2, specify the total number of isolates obtained. How many isolates were obtained from soil/rhizosphere, etc... For example: „ A total of 11 fungal isolates (A1-A11) were obtained after 7 days incubation on PDA (Table 2).....“ Perhaps the authors should think about moving Table 2 to Materials and Methods section.

Line 319-321: The sentences can be deleted.

Line 323: Indicate that accession numbers are listed in Table 2.

Line 323-325: Explain the results observed based on the performed phylogenetic analysis and NCBI blast (specify percent identity and strains with which the tested isolates show the highest homology). A new phylogenetic tree, with more reference strains should be constructed. Specify were all tested isolates homogenous among themselves or were they separated into groups, etc.

Figure 2 –title: Maximum likelihood phylogenetic tree based on concatenated ITS and β-tubulin sequences of the 11 tested isolates and ....(add number) reference T. asperellum strains retrieved from the NCBI database. The tree was rooted using C. globosum strain CBS 160.62.

Lines 378 and 382: showed the highest production

Line 397: culture filtrates

The authors should specify which seven isolates were selected in Materials and Methods section (add isolates codes).

Line 399-400: Ethyl acetate concentrates were subjected to GC-MS analysis to identify their constituents.“ The sentence can be deleted

Line 400: “The number of antimicrobial compounds was maximum” – rephrase

Line 429: Please refer to the Table 5.

Line 446: Explain the abbreviation VOCs upon the first use

Line 470-471: The sentence can be deleted.

Line 481-482: The sentence can be moved to Discussion section.

Discussion

General comment: Discussion section should be significantly improved and enlarged. Likewise written they are just a repetition of the results. Each result and compound detected should be properly discussed and some conclusions should be drawn.

Lines 497-506: References are missing

Line 510-511: were also confirmed in the previous reports

Line 512: molecular strategies are an ideal tool

Line 517: Does the sentence refer to the isolates from this work? If so, what are the references for?

Author Response

Introduction

Line 45-46: The sentence can be deleted

Yes, we corrected in the Manuscript

Line 55-57: Add reference

Added in the manuscript

Line 62: delete “fungus species in the hypocreaceae family”

Yes, we corrected in the Manuscript

Line 61-63: Add reference

Added in the manuscript

Line 64: Arabidopsis

Corrected in the Manuscript

Line 68: delete “induced”

Corrected in the Manuscript

Line 67-68: The sentence seems similar to the second part of the previous sentence: “ISR induced by cell-free culture filtrates combines the SA and JA (Jasmonic Acid)/ET (Ethylene) signaling pathways” Please correct

Corrected in the manuscript

Line 72: Rhizoctonia solani

Corrected in the Manuscript

Line 73-77: The sentence is too long. Please rephrase.

Corrected in the Manuscript

Materials and Methods

Line 88: Rhizosphere soil of which plant(s)?

Corrected in the Manuscript

Line 90-95: The text can be deleted and rephrased as follows: “Isolation of the potential fungal biocontrol isolates from soil samples was performed on Potato Dextrose Agar (PDA) according to the procedure described by ....(add reference). Plates were incubated at 28±2°C for 5-7 days.”

Corrected in the Manuscript

Line 97: Add the number of the obtained fungal isolates. What criterion did you use to select isolates for further work?

Included in the manuscript

(20 isolated were identified by morphologically and molecularly, all the isolates were usually vigorous in growth, to test the efficacy we have carried all 20 isolates to further work)

Line 97: obtained isolates

Corrected in the Manuscript

Line 98-101: Rephrase the sentence as follows: “A highly virulent isolate S. rolfsii Sr38, obtained during our previous work (ITCC Code: 8665), was used to evaluate biocontrol potential of the obtained fungal isolates.” Add reference for S. rolfsii Sr38 if it is already published.

Corrected in the Manuscript,

Reference, we have not included because it is not yet published

Line 103-107: Rephrase the text as follows: “Morphological identification of the potential biocontrol fungal isolates was performed under a compound microscope (add manufacturer). Isolates were identified based on conidiophores branching, the shape and size of the phialides, and the conidia.”

Corrected in the Manuscript

Line 105-107: The sentences can be deleted

Corrected in the Manuscript

Line 109-112: The text can be deleted or moved to Introduction or Discussion section.

Corrected in the Manuscript

Line 113-114: “They were molecularly confirmed through the amplification of the 113 ITS1/4 and β-tubulin regions.” The sentence can be deleted.

Corrected in the Manuscript

Line 114-116: Rephrase the sentence as follows: “DNA from all (add number) isolates was extracted using a CTAB (CetylTrimethylAmmonium Bromide) procedure given by Cullings et al. [14].”

Corrected in the Manuscript

Line 116: by amplification

Corrected in the Manuscript

Line 121: consisted of

Corrected in the Manuscript

Line 124: delete “template was used for the amplification”

Corrected in the Manuscript

Line 135: confirm the species identity

Corrected in the Manuscript

Delete the sentence: “We proceeded with all the isolates for further evaluation.”

Corrected in the Manuscript

Line 136: Rephrase the sentence as follows: “The maximum likelihood phylogenetic tree was constructed in MEGA11 software based on the concatenated sequences of ITS and β-tubulin.”

Corrected in the Manuscript

Specify the final sequence lenghts (nt) of ITS and β-tubulin sequences used for the phylogenetic analysis.

Included in the manuscript

Line 138: Include more T. asperellum strains in the phylogenetic analysis

Included in the manuscript

Line 146: PDA is enough. You already explained the abbreviation.

Corrected in the Manuscript

Line 163: on one-month-old tomato plants

Corrected in the Manuscript

Line 209-216: If the assay was performed according to the completely same procedure as described in reference No. 43, there is no need to write the complete procedure. If there are some modifications from the original (cited) protocol, please specify. The same comments stands for subsections 2.6.2.-2.6.4.

The assay was performed using the reference protocol with a few changes. So, we have explained the complete procedure

Line 247: analyze?

Sentence corrected in the manuscript

Line 248-250: cultured in PDB. Delete “500 ml Erlenmeyer flask that contained” and “which had 249 previously been autoclaved at 121°C for 40 min”

Corrected in the Manuscript

Explain the abbreviation PDB upon the first use.

Corrected in the Manuscript

Line 258: Explain the abbreviation GCMS upon the first use.

Lines 261-262: Rephrase the sentence as follows: “Volatile organic compounds produced by each of the seven tested isolates were analysed on 6850C Agilent GC-MS (Agilent Technologies®, USA).”

Corrected in the Manuscript

Line 278-279: The sentence can be deleted.

Corrected in the Manuscript

Line 277: Thin Layer Chromatography assay (TLC)

Corrected in the Manuscript

Results

Line 301: “All the isolates grew at a faster rate on the PDA medium“ – explain

Modified the sentence

Line 300-305: When reffering to the Table 2, specify the total number of isolates obtained. How many isolates were obtained from soil/rhizosphere, etc... For example: „ A total of 11 fungal isolates (A1-A11) were obtained after 7 days incubation on PDA (Table 2).....“ Perhaps the authors should think about moving Table 2 to Materials and Methods section.

We have corrected the table title by incorporating the changes.

Since, the table doesn't include only the isolation details, it also includes the accession number and colony characteristics of all the 20 isolates, we kept in the result part.

It will be repeated if we move to the materials and method portion.

Line 319-321: The sentences can be deleted.

Corrected in the Manuscript

Line 323: Indicate that accession numbers are listed in Table 2.

Corrected in the Manuscript

Line 323-325: Explain the results observed based on the performed phylogenetic analysis and NCBI blast (specify percent identity and strains with which the tested isolates show the highest homology). A new phylogenetic tree, with more reference strains should be constructed. Specify were all tested isolates homogenous among themselves or were they separated into groups, etc.

Corrected in the manuscript

Figure 2 –title: Maximum likelihood phylogenetic tree based on concatenated ITS and β-tubulin sequences of the 11 tested isolates and ....(add number) reference T. asperellum strains retrieved from the NCBI database. The tree was rooted using C. globosum strain CBS 160.62.

Corrected in the manuscript

Lines 378 and 382: showed the highest production

Corrected in the Manuscript

Line 397: culture filtrates

Corrected in the Manuscript

The authors should specify which seven isolates were selected in Materials and Methods section (add isolates codes).

Corrected in the Manuscript

Line 399-400: “Ethyl acetate concentrates were subjected to GC-MS analysis to identify their constituents.“ The sentence can be deleted

Corrected in the Manuscript

Line 400: “The number of antimicrobial compounds was maximum” – rephrase

Corrected in the Manuscript

Line 429: Please refer to the Table 5.

Corrected in the Manuscript

Line 446: Explain the abbreviation VOCs upon the first use

Corrected in the Manuscript

Line 470-471: The sentence can be deleted.

Corrected in the Manuscript

Line 481-482: The sentence can be moved to Discussion section.

Corrected in the Manuscript

Discussion

General comment: Discussion section should be significantly improved and enlarged. Likewise written they are just a repetition of the results. Each result and compound detected should be properly discussed and some conclusions should be drawn.

Corrected in the manuscript

Lines 497-506: References are missing

Included in the manuscript

Line 510-511: were also confirmed in the previous reports

Corrected in the Manuscript

Line 512: molecular strategies are an ideal tool

Corrected in the Manuscript

Line 517: Does the sentence refer to the isolates from this work? If so, what are the references for?

Yes, we discussed the results obtained with this study with the previous reports.

References are deleted from that part

Reviewer 3 Report

The manuscript from Shanmugaraj et al. describes several strains of T. asperellum and their abilities to inhibit the pathogenesis caused by S. rolfsii in tomato plants. It is a beautiful report and certainly deserves publication. I am sending some corrections, comments, and suggestions below.

Lines 30-1 - It os meaningless to sum up these hydrolytic activities. Please remove these total values and mention the more relevant specific acitivites.

Line 49 - remove "worldwide"

Line 50 - remove "associated with this pathogen"

Line 72 - remove "these"

Line 75 - remove "were"

Lines 75-6 - "...[63]. Its most...

Lines 76-7 - remove "produced by T. asperellum isolates"

Line 81 - "...evaluate T. asperellum..."

Line 83-4 - "...produced by this bio-control..."

Lines 88-90 - include GPS coordinates for the soil collections

Line 92 - 9 mL

Line 93 - mL

Line 94 - 1 mL

Lines 109-114 - this is not methodology. Remove these first two sentences.

Line 114 - what is the "modified" protocol? Please describe it, or, if its fully described in Ref. [14], rephrase.

Line 121 - consisted of 12.5uL

Line 123 - 9.5 uL...1 uL...

Line 124 - 1 uL (100 ng/uL)

Line 131 - 1.2 % (w/v)...0.5 mg/uL...

Line 134 - please detail "outsorcing". It was a private service? Public platform? Which technology was used for sequencing? Which type of blast was used, against which database?

Line 167 - "The test pathogen...on sorghum grains."

Line 202 - "...and were subtracted..."

Line 203 - "...Table 1. The amount..."

Line 206 - Provide supplier and product number for all substrates. Concentrations in the table are % (w/v).

Line 209 - 1 mL of 0.5 % (w/v)

Line 210 - 1 mL

Line 211 - 15 mL

Line 212 - 3 mL

Line 218 - 0.5 mL...3.2 mg/mL...1 mL...

Lines 219-20 - "...0.5 mL....isolates separately."

Line 221 - 2 mL

Line 224 - 1 mL....2 mL....1 mL

Line 225 - 1 % (w/v) 

Line 226 - 4 mL

Line 230 - 1 mL

Line 231 - mL

Line 231 - please provide supplier, source, and product number of chitin, and protocol for preparation of colloidal chitin. 

Line 234 - 3 mL

Line 239 - 1 % (w/v)

Lines 240-2 - Please rewrite. The order of procedures and incubations is not clear.

Line 249 - 500 mL....100 mL

Line 263 - components were

Line 265 - 1 mL/min

Line 266 - 1 uL

Line 283 - please give brand and product number of silica TLC plates

Line 285 - please describe the two solvents systems here as well

Line 301 - please rewrite "at a faster rate". Was this is overnight? 2-3 days? 1 week? Please be more specific.

Line 306 - Table 2 describes much more than morphological characteristics. Please include the other parameters in the Table with proper explanation and abbreviation meaning as well.

Line 320 - ...ITS and beta-tubulin...

Line 322 - please describe outsourcing (where and how).

Line 327 - Figure 2. I see no meaning in this tree. It just shows that the isolates are closer to T. asperellum than to C. globosum. There is no description about the proximity of isolates among themselves or internal clades. I see three options here: include better description and discussion of phylogenetics relationships between isolates, or build a new tree with other relevant close species (which would be a lot more meaningful), or just remove the Figure.

Line 343 - Table 3. Remove the "Mean" column. There is no sense in averaging such different techniques, they have completely different meanings. Besides that, please include errors/deviations for each measurement in the Table, and number of replicates in the Legend. What is CD@5%? SEM was calculated with all the data? Please remove and include deviations for each particular measurement. Describe which statistical test was used for comparison between groups in the Legend.

Line 347 - Figure 3. The numbers/letters for each plate are too small, please double the size. In the legend: "...isolates on S. rolfsii growth..."

Line 356-7 - This inhibition is complementary to disease incidence. Remove this sentence.

Lines 360-2 - Same as above. Remove sentence.

Lines 364-5 - same as above, remove.

Line 366 - Remove ROC columns, they are complementary to DSI and redundant info. Include number of replicas, error/deviation for all measurements, and description of the test. Remove CD@5% and SEM lines.

Line 373 - Figure 4. Figures 4A, 4B, and 4C are poorly illustrative and must be removed. This will be give emphasis to Figure 4D, which is the more relevant part.

Line 378 - ...produced the highest amount... 

Line 379 - ...and the least in A15...

Lines 382-3 - ...the highest amount of cellulase...

Lines 384-5 - ...(106.06 IU/mg) showed the highest production...

Line 387 - ...the maximum yield, followed...

 Lines 388-90 - Remove this sentence. There is no meaning in suming up activities that are completely different.

Line 394 - Table 5. Remove the column "Total" (reason above). Include number of replicates, errors, and test used.

Line 405-6 - ...A17 showed many metabolites,...

Line 410 - ...respectively, were recorded...

Lines 414-5 - ...isolate do not produce many bioactive...

Line 419 - Table 5.  Include number of replicates, errors are SEM or SD?

Line 438 - ....observed several hydrocarbons...

Lines 441-2 - ...(4.07%) were observed at the highest content in the isolate A17.

Line 445-6 - described the patterns that can be observed in the heatmap, otherwise remove.

Line 479 - rewrite: what do you mean with "more potential isolates"? potent? promising? Please rephrase.

Lines 481-2 - "...correlation with metabolite..."

Line 501 - "...pathogen. Recently, the use..."

Line 505 - "...study evaluated the T. ..."

Line 515 - Authors justified the use of molecular markers, but not comment or discuss the results they obtained with them. Please write a sentence or two on this.

Line 526 - remove the inhibitions, it´s repetitite

Line 532 - "...The production of hydrolytic..."

Line 538 - "...isolates. The isolate A10 showed the...

Line 539 - delete (583.18 IU/mg)

Line 545 - ...isolates selected....

Line 550 - ...having antifungal...

Line 558 - ...system is then recommended...

The English is good but needs polishing (see comments above).

Author Response

Lines 30-1 - It is meaningless to sum up these hydrolytic activities. Please remove these total values and mention the more relevant specific activities.

Corrected in the manuscript

Line 49 - remove "worldwide"

Corrected in the manuscript

Line 50 - remove "associated with this pathogen"

Corrected in the manuscript

Line 72 - remove "these"

Corrected in the manuscript

Line 75 - remove "were"

Corrected in the manuscript

Lines 75-6 - "...[63]. Its most...

Corrected in the manuscript

Lines 76-7 - remove "produced by T. asperellum isolates"

Corrected in the manuscript

Line 81 - "...evaluate T. asperellum..."

Corrected in the manuscript

Line 83-4 - "...produced by this bio-control..."

Corrected in the manuscript

Lines 88-90 - include GPS coordinates for the soil collections

Added in the table 2

Line 92 - 9 mL

Corrected in the manuscript

Line 93 - mL

Corrected in the manuscript

Line 94 - 1 mL

Corrected in the manuscript

Lines 109-114 - this is not methodology. Remove these first two sentences.

Corrected in the manuscript

Line 114 - what is the "modified" protocol? Please describe it, or, if its fully described in Ref. [14], rephrase.

Corrected in the manuscript

Line 121 - consisted of 12.5uL

Corrected in the manuscript

Line 123 - 9.5 uL...1 uL...

Corrected in the manuscript

Line 124 - 1 uL (100 ng/uL)

Corrected in the manuscript

Line 131 - 1.2 % (w/v)...0.5 mg/uL...

Corrected in the manuscript

Line 134 - please detail "outsorcing". It was a private service? Public platform? Which technology was used for sequencing? Which type of blast was used, against which database?

Included in the manuscript

Line 167 - "The test pathogen...on sorghum grains."

Corrected in the manuscript

Line 202 - "...and were subtracted..."

Corrected in the manuscript

Line 203 - "...Table 1. The amount..."

Corrected in the manuscript

Line 206 - Provide supplier and product number for all substrates. Concentrations in the table are % (w/v).

Included in the manuscript

Yes, the concentrations in the table are % (w/v)

Line 209 - 1 mL of 0.5 % (w/v)

Corrected in the manuscript

Line 210 - 1 mL

Corrected in the manuscript

Line 211 - 15 mL

Corrected in the manuscript

Line 212 - 3 mL

Corrected in the manuscript

Line 218 - 0.5 mL...3.2 mg/mL...1 mL...

Corrected in the manuscript

Lines 219-20 - "...0.5 mL....isolates separately."

Corrected in the manuscript

Line 221 - 2 mL

Corrected in the manuscript

Line 224 - 1 mL....2 mL....1 mL

Corrected in the manuscript

Line 225 - 1 % (w/v) 

Corrected in the manuscript

Line 226 - 4 mL

Corrected in the manuscript

Line 230 - 1 mL

Corrected in the manuscript

Line 231 - mL

Corrected in the manuscript

Line 231 - please provide supplier, source, and product number of chitin, and protocol for preparation of colloidal chitin. 

Included in the manuscript

Line 234 - 3 mL

Corrected in the manuscript

Line 239 - 1 % (w/v)

Corrected in the manuscript

Lines 240-2 - Please rewrite. The order of procedures and incubations is not clear.

Corrected in the manuscript

Line 249 - 500 mL....100 mL

Corrected in the manuscript

Line 263 - components were

Corrected in the manuscript

Line 265 - 1 mL/min

Corrected in the manuscript

Line 266 - 1 uL

Corrected in the manuscript

Line 283 - please give brand and product number of silica TLC plates

Included in the manuscript

Line 285 - please describe the two solvents systems here as well

Included in the manuscript

Line 301 - please rewrite "at a faster rate". Was this is overnight? 2-3 days? 1 week? Please be more specific.

Corrected in the manuscript

Line 306 - Table 2 describes much more than morphological characteristics. Please include the other parameters in the Table with proper explanation and abbreviation meaning as well.

Revised in the manuscript

(GPS included, title changed)

Line 320 - ...ITS and beta-tubulin...

Corrected in the manuscript

Line 322 - please describe outsourcing (where and how).

Corrected in the manuscript

Line 327 - Figure 2. I see no meaning in this tree. It just shows that the isolates are closer to T. asperellum than to C. globosum. There is no description about the proximity of isolates among themselves or internal clades. I see three options here: include better description and discussion of phylogenetics relationships between isolates, or build a new tree with other relevant close species (which would be a lot more meaningful), or just remove the Figure.

Corrected in the manuscript

Line 343 - Table 3. Remove the "Mean" column. There is no sense in averaging such different techniques, they have completely different meanings. Besides that, please include errors/deviations for each measurement in the Table, and number of replicates in the Legend. What is CD@5%? SEM was calculated with all the data? Please remove and include deviations for each particular measurement. Describe which statistical test was used for comparison between groups in the Legend.

Corrected in the manuscript

Critical difference @ 5% of significance

Yes, SEm was calculated with all the data.

Included the standard deviation for each measurement. Duncan’s multiple range test was test was used for the comparison.

Line 347 - Figure 3. The numbers/letters for each plate are too small, please double the size. In the legend: "...isolates on S. rolfsii growth..."

Corrected in the manuscript

Line 356-7 - This inhibition is complementary to disease incidence. Remove this sentence.

Corrected in the manuscript

Lines 360-2 - Same as above. Remove sentence.

Corrected in the manuscript

Lines 364-5 - same as above, remove.

Corrected in the manuscript

Line 366 - Remove ROC columns, they are complementary to DSI and redundant info. Include number of replicas, error/deviation for all measurements, and description of the test. Remove CD@5% and SEM lines.

Corrected in the manuscript

Line 373 - Figure 4. Figures 4A, 4B, and 4C are poorly illustrative and must be removed. This will be give emphasis to Figure 4D, which is the more relevant part.

Corrected in the manuscript

Line 378 - ...produced the highest amount... 

Corrected in the manuscript

Line 379 - ...and the least in A15...

Corrected in the manuscript

Lines 382-3 - ...the highest amount of cellulase...

Corrected in the manuscript

Lines 384-5 - ...(106.06 IU/mg) showed the highest production...

Corrected in the manuscript

Line 387 - ...the maximum yield, followed...

Corrected in the manuscript

 Lines 388-90 - Remove this sentence. There is no meaning in suming up activities that are completely different.

Corrected in the manuscript

Line 394 - Table 5. Remove the column "Total" (reason above). Include number of replicates, errors, and test used.

Corrected in the manuscript.

Line 405-6 - ...A17 showed many metabolites,...

Corrected in the manuscript

Line 410 - ...respectively, were recorded...

Corrected in the manuscript

Lines 414-5 - ...isolate do not produce many bioactive...

Corrected in the manuscript

Line 419 - Table 5.  Include number of replicates, errors are SEM or SD?

Errors are SEm

Line 438 - ....observed several hydrocarbons...

Corrected in the manuscript

Lines 441-2 - ...(4.07%) were observed at the highest content in the isolate A17.

Corrected in the manuscript

Line 445-6 - described the patterns that can be observed in the heatmap, otherwise remove.

Included in the manuscript

Line 479 - rewrite: what do you mean with "more potential isolates"? potent? promising? Please rephrase.

Corrected in the manuscript

Lines 481-2 - "...correlation with metabolite..."

Corrected in the manuscript

Line 501 - "...pathogen. Recently, the use..."

Corrected in the manuscript

Line 505 - "...study evaluated the T. ..."

Corrected in the manuscript

Line 515 - Authors justified the use of molecular markers, but not comment or discuss the results they obtained with them. Please write a sentence or two on this.

Included in the manuscript

Line 526 - remove the inhibitions, it´s repetitite

Corrected in the manuscript

Line 532 - "...The production of hydrolytic..."

Corrected in the manuscript

Line 538 - "...isolates. The isolate A10 showed the...

Corrected in the manuscript

Line 539 - delete (583.18 IU/mg)

Corrected in the manuscript

Line 545 - ...isolates selected....

Corrected in the manuscript

Line 550 - ...having antifungal...

Corrected in the manuscript

Line 558 - ...system is then recommended...

Corrected in the manuscript
